# Deciphering the unusual fluorescence in weakly coupled bis-nitro-pyrrolo[3,2-*b*]pyrroles

Yevgen M. Poronik[1], Glib V. Baryshnikov [2], Irena Deperasińska[3], Eli M. Espinoza [4,9], John A. Clark[5], Hans Ågren [2,6✉], Daniel T. Gryko [1✉] & Valentine I. Vullev [4,5,7,8✉]

Electron-deficient π-conjugated functional dyes lie at the heart of organic optoelectronics. Adding nitro groups to aromatic compounds usually quenches their fluorescence via intersystem crossing (ISC) or internal conversion (IC). While strong electronic coupling of the nitro groups with the dyes ensures the benefits from these electron-withdrawing substituents, it also leads to fluorescence quenching. Here, we demonstrate how such electronic coupling affects the photophysics of acceptor–donor–acceptor fluorescent dyes, with nitrophenyl acceptors and a pyrrolo[3,2-*b*]pyrrole donor. The position of the nitro groups and the donor-acceptor distance strongly affect the fluorescence properties of the *bis*-nitrotetraphenylpyrrolopyrroles. Concurrently, increasing solvent polarity quenches the emission that recovers upon solidifying the media. Intramolecular charge transfer (CT) and molecular dynamics, therefore, govern the fluorescence of these nitro-aromatics. While balanced donor-acceptor coupling ensures fast radiative deactivation and slow ISC essential for large fluorescence quantum yields, vibronic borrowing accounts for medium dependent IC via back CT. These mechanistic paradigms set important design principles for molecular photonics and electronics.

[1] Institute of Organic Chemistry, Polish Academy of Sciences, Warsaw, Poland. [2] Department of Physics and Astronomy, Uppsala University, Uppsala, Sweden. [3] Institute of Physics, Polish Academy of Sciences, Warsaw, Poland. [4] Department of Chemistry, University of California, Riverside, CA, USA. [5] Department of Bioengineering, University of California, Riverside, CA, USA. [6] Department of Physics, Tomsk State University, 36 Lenin Avenue, Tomsk 634050, Russian Federation. [7] Department of Biochemistry, University of California, Riverside, CA, USA. [8] Materials Science and Engineering Program, University of California, Riverside, CA, USA. [9] Present address: College of Bioengineering, University of California, Berkeley, CA 94720, USA. ✉email: hagren@kth.se; dtgryko@icho.edu.pl; vullev@ucr.edu

Electron-deficient aromatic π-conjugated molecules are essential for the design of organic analogs of *n*-doped semiconductors[1], and can serve as light sensitizers for photooxidation and for initiating charge transfer (CT) via hole transduction[2]. Placing electron-withdrawing substituents or positive charges on aromatic rings constitute the most common means for decreasing the electron density. With Hammett constants exceeding 0.7, the nitro group is one of the strongest electron-withdrawing moieties[3]. With a few exceptions, however, nitro-aromatics are not fluorescent[4], which renders them impractical for a wide variety of photonic and optoelectronic applications where long-lived singlet-excited states are mandatory.

Three mechanisms define the modes by which nitro groups quench fluorescence. (1) Nitro-aromatics have numerous closely packed π-π* and n-π* electronic excited states with different multiplicities that favor intersystem crossing (ISC) leading to efficient triplet formation[5,6]. (2) The strongly electron-withdrawing nitro groups can favor the formation of dark CT states that non-radiatively decay to the ground state *via* back charge transfer[7]. (3) The rotation around the carbon-nitrogen bond, linking the nitro group with the aromatic ring, can lead to conical intersections between the excited and the ground states providing non-radiative pathways for efficient internal conversion (IC)[8]. In cases when $NO_2$ group is highly twisted (e.g., 9-nitroanthracene), the important pathway from $S_1$ corresponds to a photochemical reaction that generates nitrogen(II) oxide (NO·)[9,10].

Electron-deficient organic chromophores with long-lived singlet excited states, i.e., exhibiting strong fluorescence, are invaluable for a range of photonic and electronic applications, such as photocatalysis and energy conversion[11]. Fluorescent nitro-aromatics, however, are few and far between. Adding $NO_2$ group(s) to even strongly fluorescent aromatic molecules quenches their emission as exemplified by dinitrofluorescein[12], nitro-pyrenes[13–15], para-nitroanilines[16,17], and 4-amino-1-nitronaphthalene[18]. Despite the acknowledged importance of excited states with a CT character, these examples illustrate the lack of a good understanding of how to unequivocally make nitro-aromatics fluoresce.

Introducing CT character in the excited states of nitro-aromatics can indeed disfavor ISC[19,20], and strongly emissive amino-derivatives of nitro-2,1,3-benzoxadiazole (NBD) benefit from that[21–23]. The same CT character of singlet excited states, however, lowers their energy levels bringing them close to the ground state that can lead to the formation of conical intersections (CIs) providing pathways for efficient IC[24].

A partial decoupling of the nitro groups from the principal π-conjugated system of the chromophore provides an alternative means for attaining electron-deficient fluorescent conjugates, i.e., conjugates that benefit from the electron-withdrawing capabilities of the nitro group without suffering from its inherent propensity for quenching their fluorescence[25,26]. Specifically, placing nitro groups on aromatic rings, connected via a single σ-bonds with the principal chromophore, allows for attaining fine balance in partial electronic coupling. Bond rotation in the excited states of such flexible structures allow for electronic decoupling with the nitro-containing electron-acceptor moieties[5]. That is, after photo-excitation such chromophores relax to twisted intramolecular charge-transfer (TICT) states, which usually do not favor ISC, allowing them to exhibit red-shifted fluorescence[27,28].

Studies of disubstituted benzenes[29], *bis*-ethynobenzenes[30], disubstituted naphthalenes[31], and fused nitro-fluorenes[32,33] demonstrate these design principles for organic branched fluorescent nitro-aromatics. In these examples, an increase in solvent polarity increases the fluorescence efficiency, which indicates that radiative deactivation occurs from states with a pronounced CT

character[8,34]. Considerably more frequently, however, such designs, involving the addition of nitro groups to aryl substituents, lead to pronounced fluorescence quenching. Conversely, placing electron-donating groups on the nitro-containing electron-acceptor aryls weakens the CT character of the excited states and leads to fluorescence enhancement[35].

Furthermore, the extra torsional degrees of freedom of flexible structures, which contain nitroaryl substituents, can facilitate additional pathways for non-radiative deactivation of the singlet excited states[36,37]. The rotation of the aromatic rings around the C–C, and of the nitro groups around the N–C, σ-bonds yields a myriad of conformers, only a handful of which may have propensity for radiative deactivation[5,8,19]. Other conformers correspond to minima on the potential-energy surface (PESs) of the lowest singlet excited state that lead to the formation of CIs mediating efficient IC.

This complexity of the modes in which the various characteristics control the dynamics of radiative and non-radiative decays has impeded the development of broadly applicable robust guidelines for designing fluorescent nitro-aromatics. It warrants aggressive improvements of the understanding how the synergy between different parameters affect the excited-state dynamics of fluorophores that bear nitro groups.

This article interrogates how the position of nitro groups on phenyl substituents, along with the distance from the principal chromophore, affects its fluorescence. We employ derivatives of tetraaryl-pyrrolo[3,2-*b*]pyrrole (TAPP) (Fig. 1), which belongs to a recently developed class of organic dyes with a great potential for photonic and electronic applications[38–41]. Recently, we discovered that 2,5-bis(4-nitrophenyl)-1,4-bis(4-octylphenyl)-1,4-dihydropyrrolo[3,2-*b*]pyrrole (**1p**, Fig. 1) exhibits strong fluorescence and huge solvatofluorochromism, which originates from its quadrupole symmetry that breaks in the excited state[42–44]. Herein, we demonstrate that moving the nitro groups of **1p** to *meta* and *ortho* positions, and adding extra phenylene linkers (Fig. 1), changes the energy levels of the CT states, as expected from the inverse relation between the CT driving forces and the donor-acceptor distance[45]. Bringing the nitro group closer to the pyrrolopyrrole core drastically decreases $\phi_f$ of the TAPPs. Concurrently, an increase in solvent polarity also quenches the fluorescence of all TAPP nitro derivatives. While these findings are consistent with CT-induced IC, they do not explain what makes such nitro-aromatics fluorescent. Further analysis reveals that in addition to affecting the energy levels of the CT states, moving the nitro group drastically changes the radiative-decay and ISC rate constants, $k_f$ and $k_{ISC}$, respectively. An increase in $k_f$ is as apparent as it is a challenging approach for improving $\phi_f$ and it has never been the focus in the designs of fluorescent nitro-aromatics. Conversely, the solvent dependence of $\phi_f$ reflects polarity-induced changes in the IC rates. The Herzberg–Teller (HT), rather than the FC, approach is essential for analyzing the $S_1{\rightarrow}S_0$ non-radiative transitions, revealing the importance of vibronic borrowing for efficient IC in such conjugates, which represents another unexplored consideration that is important for the design of fluorescent nitro-aromatics.

## Results

**Rationale for the molecular design.** While the nitro groups on the phenyl substituents of the TAPPs act as electron acceptors (A), the pyrrolopyrrole condensed-ring system is exceptionally electron rich[46] and acts as an electron donor (D) when photo-excited. We designed two sets of 2,5-bis(nitrophenyl)-TAPPs with a pseudo-$C_2$ symmetry and an A-D-A configuration (Fig. 1): **1p**, **1m** and **1o**, in which the nitro groups are at the *para*-, *meta*- and *ortho*- positions, respectively, in regard to the pyrrolo[3,2-*b*]

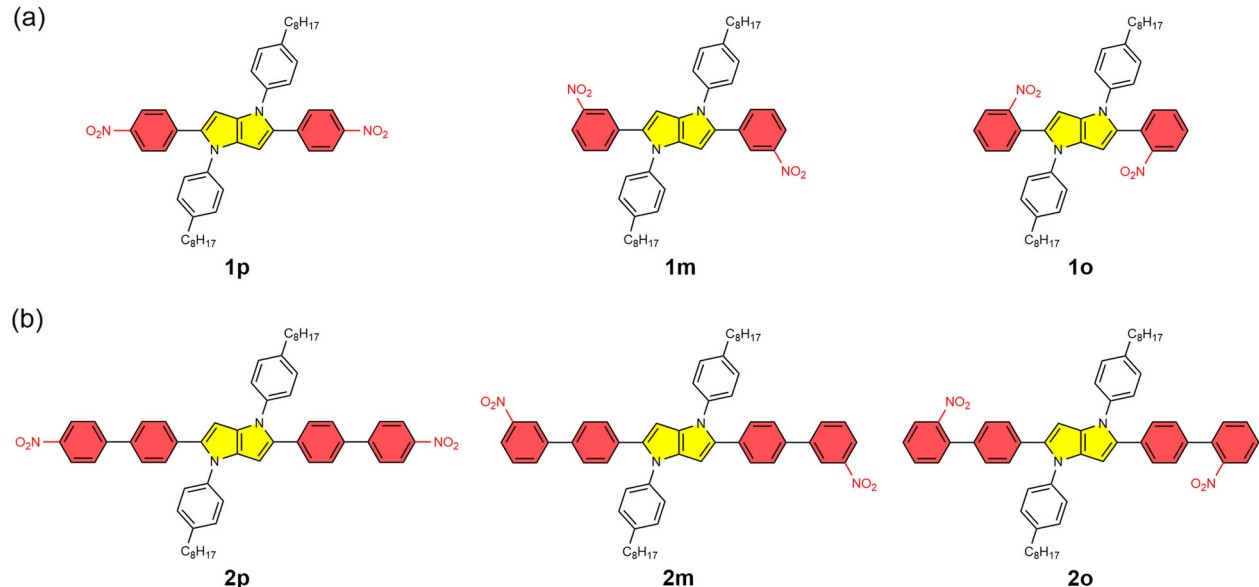

**Fig. 1 2,5-Bis(aryl)-pyrrolo[3,2-*b*]pyrroles bearing nitro groups at *para-*, *meta-*, and *ortho-* positions respectively. a** Structures of 2,5-bis(nitrophenyl)-1,4-dihydropyrrolo[3,2-*b*]pyrroles. **b** Structures of 2,5-bis(nitro-[1,1'-biphenyl]-4-yl)-1,4-dihydropyrrolo[3,2-*b*]pyrroles.

pyrrole core; and **2p**, **2m**, and **2o**, which possess extra phenylene linkers between the nitrophenyl substituents and the pyrrolo-pyrrole core. The synthesis of these dyes involves multi-component reactions between appropriate nitro-substituted aldehydes, 4-octylaniline and butano-2,3-dione[47,48] and deliver the final products in 5–40% yields (see the Supplementary Scheme 1, Supplementary Figs. 1–8).

Placing the nitro groups at *para*, *meta*, or *ortho* positions (Fig. 1a) makes it possible to examine the effects of donor-acceptor coupling on the excited-state dynamics. The electronic coupling between donors and acceptors linked with phenyl rings varies depending on their relative positions, where *ortho* and *para* configurations ensure strong electronic coupling, whereas *meta* does not[49]. While the coupling should be strong for both, **1o** and **1p**, the steric hindrance of the nitro group when it is at the *ortho* position twists the conformation and decreases the electronic coupling between the nitrobenzene moiety and the pyrrolopyrrole core. As computational analysis reveals, the nitro groups are rotated at 43° and 37° for the ground electronic states of **1o** and **2o**, respectively. Conversely, the *meta-* and *para-*substituted derivatives, **1m**, **1p**, **2m**, and **2p**, exhibit a planar arrangement of the nitro groups and the benzene ring, the dihedral angles are less than 1°. These structural features suggest further weakening of the electronic coupling with the nitro groups in the *ortho* derivatives.

**Optical absorption spectra.** The *bis*-nitro-TAPPs manifest absorption that extends to about 500 nm into the visible spectral region (Fig. 2a–h, Supplementary Table 1). Our quantum-chemical calculations reproduce the absorption spectra of the *bis*-nitro-TAPPs (Fig. 2i–k). Despite the slight systematic over-estimation of the transition energies, the computational analysis provides an assignment for the optical transitions. For all TAPPs the first observed absorption band corresponds to the $S_0 \rightarrow S_1$ electronic transition accompanied by predominant HOMO-LUMO single-electron excitation of charge-transfer nature. In all cases, HOMO is localized on a central pyrrolo[3,2-*b*]pyrrole donor fragment, while LUMO is equally localized on the edge nitrophenyl groups (Supplementary Figs. 9, 10 and in Supplementary Data 1–6). Thus, one can assign $S_1$ electronic state of TAPPs. (Supplementary Figs. 9–12 and in Supplementary Data files 1–6).

The absorption maxima of the TAPPs exhibit minute dependence on medium polarity (Fig. 3a), which is consistent with the quadrupole symmetry of their ground and LE states. Between 300 and 550 nm, the spectra of **1p**, **1m,** and **1o** show a UV and a visible band (Fig. 2a–c, Supplementary Table 2). For **1p**, the visible band is at about 470 nm and dominates the spectrum, while the UV one appears as a hypsochromic shoulder (Fig. 2a). For **1o**, the two bands are distinctly separated with the UV peak about four times more intense than the broad visible one (Fig. 2c). The spectra of **1m** also show the prevalence of the UV signal, while the visible band is shifted to the blue and appears as a bathochromic shoulder (Fig. 2b). In contrast, single broad bands compose the spectra of **2p**, **2m,** and **2o** (Fig. 2d–f), which can originate from overlapping of multiple transitions resultant from the increased conformational degrees of freedom from the additional phenylenes.

The spectral features of **1o** resemble those of **1m**, rather than of **1p**. The steric hindrance from the nitro group at the *ortho* position most likely suppresses the establishment of planarity between the pyrrolo[3,2-*b*]pyrrole and the benzene ring, compromising the through-bond electronic coupling between them. That is, while in **1m** the electronic coupling is weakened within the benzene, in **1o** it is weakened at the bond between the nitrophenyls and the heterocyclic core. Concurrently, theoretical calculations reveal a strong correlation between the mutual rotation degree of benzene ring relative to the adjacent pyrrolo[3,2-*b*]pyrrole (54.5°, 38.0°, 36.0° for **1o**, **1m**, **1p**, respectively) and C–C bond length between these moieties (1.470 Å, 1.466 Å, 1.462 Å for **1o**, **1m**, **1p**, respectively).

Actually, the larger twisting between pyrrolo[3,2-*b*]pyrrole and the benzene ring the less efficient is through-bond electronic coupling between them. Therefore, **1p**, **1m** and **1o** exhibit different spectral features originating from three distinctly different patterns of electronic coupling between the pyrrolopyr-role core and the nitro groups.

When the nitro groups are at the *para* positions, i.e., in **1p** and **2p**, the absorption maxima are situated between about 420 and 480 nm (Fig. 2a, d). The sharp absorption bands in the visible spectral region is consistent with strong electronic coupling between the electron-rich core and the nitrophenyl substituents containing a *para* $NO_2$ group (Figs. 1 and 2a, d). Adding the extra

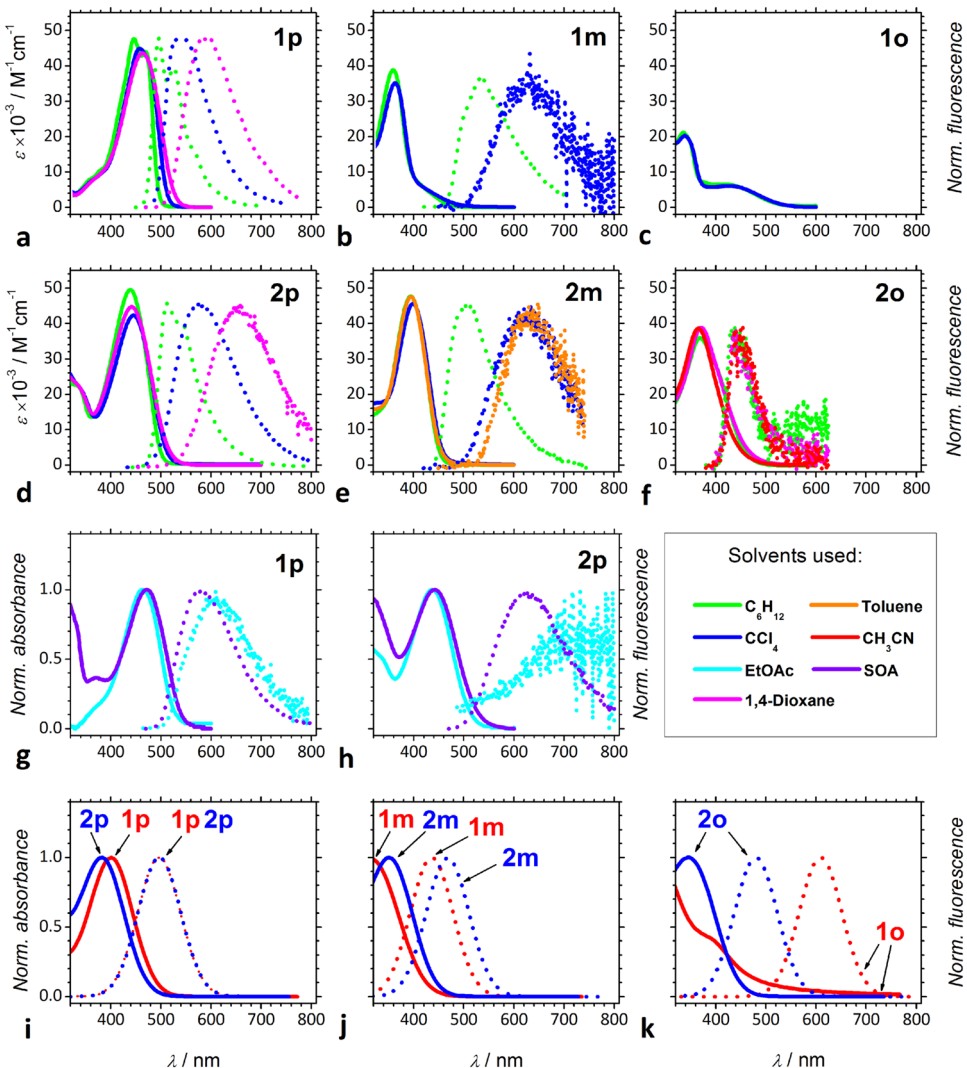

**Fig. 2 Absorption and normalized fluorescence spectra of the *bis*-nitro-TAPPs. a–f** experimentally recorded for solvents with different polarity; **g, h** experimentally recorded for sucrose octaacetate (SOA) vs. ethyl acetate (EtOAc); and **i–k** calculated for the gas phase, using TD-DFT. The solid lines represent the absorption spectra; and the dotted lines—the fluorescence spectra. For **1p**, $\lambda_{ex} = 460$ nm for $C_6H_{12}$ and $CCl_4$, and $\lambda_{ex} = 464$ nm for the rest of the solvents; for **1m**, $\lambda_{ex} = 347$ nm; for **2p**, $\lambda_{ex} = 440$ nm; and for **2m** and **2o**, $\lambda_{ex} = 373$ nm.

phenylene linkers that extends the π-conjugation causes 20-nm hypsochromic shift in the absorption maximum of **2p** in comparison with **1p** (Figs. 2a, e, 3a). This finding suggests that the extra phenylenes in **2p** stabilize its grounds state more than its LE state.

Conversely, **1m**, **1o**, **2m**, and **2o** manifest sharp absorption bands with maxima located in the UV spectral region (Fig. 2b, c, e, f). The relatively weak absorption at the long-wavelength edge of the spectra of **1m** barely extends to about 460 nm (Fig. 2b), which is consistent with the inherently weakened electronic coupling with the *meta* nitro groups. The additional phenylene spacers in **2m** result in spectral features that are intermediate between those of **1p** and **1m**. That is, the phenylene-induced extension of the π-conjugation in **2m** causes an approximately 35-nm bathochromic shift of the sharp UV peak observed for **1m**. Concurrently, the shoulder extending the absorption of **1m** into the visible region is not apparent in the spectra of **2m**. The phenylene spacers further reduce the relatively weak electronic coupling with the *meta* nitro groups. This decrease in the electronic coupling lowers the amplitude of the broad shoulder apparent at the long-wavelength edges of the spectra of **1m**, making it unnoticeable in the **2m** absorption (Fig. 2b, e).

The addition of phenylene spacers causes changes in the absorption spectra of **1o** similar to those we observe for the *meta* analogs, **1m** and **2m**. Specifically, **2o** exhibits a UV absorption band with a maximum at about 370 nm that is: (1) 10–40 nm red-shifted in comparison with the UV absorption peaks of **1o** and **1m**; and (2) 50–60 nm blue-shifted in comparison with the broad absorption band of **1o** in the visible spectral region. Furthermore, the broad visible band in the absorption spectra of **1o**, extending to about 540 nm, is not noticeable in the spectra of **2o** (Fig. 2c, f). The bathochromic shift of the UV band upon addition of phenylene linkers is consistent with extension of the π-conjugation over the additional phenylenes (Fig. 2c, f). The mutual rotation between the pyrrolo[3,2-*b*]pyrrole core and adjacent benzene rings is similar for **2o**, **2m**, and **2p** molecules (35.1°, 34.3°, 33.7°, respectively) that is also similar to the rotation between benzene rings in **2p** molecule (36.1°). This means a full equalization of π-conjugation in **2p** molecule and general extension of the π-conjugation over the additional phenylenes and central pyrrolo[3,2-*b*]pyrrole core for a whole series of **2o**, **2m**, **2p** TAPPs. The disappearance of the visible band in the spectrum of **2o** suggests for further weakening of the electronic coupling of the nitro groups with the pyrrolopyrrole core,

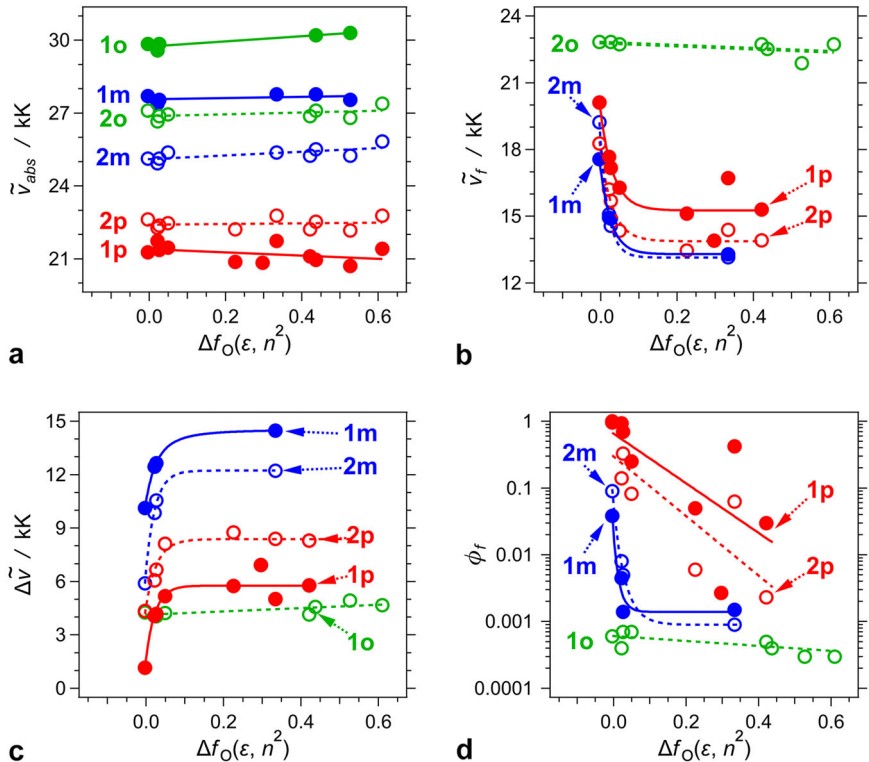

**Fig. 3 Dependence of photophysical properties of the _bis_-nitro-TAPPs on solvent polarity.** Onsager solvation function $\Delta f_O(\varepsilon, n^2) = f_O(\varepsilon) - f_O(n^2)$, where $f_O(x) = 2(x - 1)/(2x + 1)$, accounts for the orientational and nuclear solvent polarization. Solvents used (with the corresponding values of their $\Delta f_O(\varepsilon, n^2)$ shown in parentheses): cyclohexane ($-0.0036$), $CCl_4$ (0.022), toluene (0.026), 1,4-dioxane (0.049), anisole (0.23), chloroform (0.30), _t_-butyl methyl ether (0.33), THF (0.42), DCM (0.44), DMSO (0.53), and acetonitrile (0.61).

induced by the linkage phenylenes. The broad absorption bands of **2o** and **2m** (Fig. 2e, f), extending to 500 nm, are comparable with the spectral features of TAPPs bearing moderately strong electron-withdrawing groups[47,50] or biphenyl substituents[51].

**Optical emission.** Apart from **1o**, _bis_-nitro-TAPPs fluoresce (Fig. 2, Table 1). The emission spectra of the fluorescent _bis_-nitro-TAPPs, except of dye **2o**, manifest strong positive solvatochromism (Figs. 2 and 3b, c). This pronounced solvato-fluorochromism is consistent with the breaking of the quadrupolar symmetry of the FC LE states leading to dipolar $S_1$ states capable of radiative deactivation[37,52–56].

While an increase in solvent polarity causes bathochromic shifts in the fluorescence of **1p** and **2p**, the emission maxima of **2p** are consistently at longer wavelengths than those of **1p** (Figs. 2, 3). Similar to the _para_-nitro TAPPs, **2p** and **1p**, the fluorescence spectral maxima of **2m** are also consistently bathochromically shifted compared to those of **1m** (Fig. 2, Table 1). The larger Stokes shifts of **2p** and **2m**, in comparison with those of **1p** and **1m**, respectively, are consistent with (1) an increase in the energy differences between the FC and relaxed geometries of the ground and excited states, induced by the extra conformational degrees of freedom from the extra two pheny-lenes in **2p** and **2m**; and (2) an increase in the distance between the nitrophenyl acceptors and the pyrrolopyrrole donor, which enhances the dipole of the excited states with a CT character.

In contrast to the other fluorescent _bis_-nitro-TAPPs, **2o** shows no solvatofluorochromism (Fig. 2f). Its fluorescence maximum is at about 440 nm and does not exhibit strong dependence on the solvent polarity (Fig. 3b). This finding suggests that solvent polarity affects similarly the singlet excited and ground states involved in the radiative deactivation of **2o**.

The dependence of the Stokes' shift, $\Delta \tilde{\nu}$, on the Onsager polarity, $\Delta f_O(\varepsilon, n_D^2)$, reveals the difference between the electric dipole moments, $\boldsymbol{\mu}_0$ and $\boldsymbol{\mu}^*$, of the ground and the fluorescent excited state, respectively, for species with radius $r$, as implemented by the Lippert–Mataga–Ooshika formalism (Fig. 3c)[57–60]:

$$\Delta \tilde{\nu} = \frac{\left(\boldsymbol{\mu}^* - \boldsymbol{\mu}_0\right)^2}{2\,\varepsilon_0\,h\,c\,r^3}\,\Delta f_O\left(\varepsilon,\,n^2\right) + \Delta \tilde{\nu}_0 \tag{1}$$

where $\Delta \tilde{\nu}_0$ is the Stokes' shift for non-polar media with $\Delta f_O(\varepsilon, n^2) = 0$.

The Stokes' shifts of the _para_ and _meta bis_-nitro-TAPPs manifest a saturation type of dependence on solvent polarity (Fig. 3c). This behaviour is consistent with transitions to non-emissive excited states that dominate in polar media. Increasing solvent polarity not only enhances $\Delta \tilde{\nu}$, but also quenches the fluorescence. That is, the increase in $\Delta \tilde{\nu}$ is consistent with bringing the energy levels of the excited closer to the PES of $S_0$, and enhancing non-radiative deactivation rates. Thus, the detected weak emission for solvents with moderate polarity originates from minute populations of excited-state conformers with Stokes' shifts that are relatively small to ensure fluorescence. To prevent increases in $\Delta \tilde{\nu}$ for polar media and detectable radiative deactivation, the dipole moments, $\boldsymbol{\mu}^*$, of these fluorescent conformers should be similar to $\boldsymbol{\mu}_0$ of $S_0$. These results are consistent with an inherent shortcoming of the Lippert–Mataga–Ooshika analysis, which assumes that the measured steady-state emission spectra, needed for calculating $\Delta \tilde{\nu}$, are representative of the whole excited-state population.

Nevertheless, focusing on the dependence of $\Delta \tilde{\nu}$ on $\Delta f_O(\varepsilon, n^2)$ for non-polar media, i.e., $\Delta f_O(\varepsilon, n^2) \lesssim 0.05$, shows a large increase in the excited-state dipole moments for the _meta_ than the _para_

**Table 1 Photophysical characteristics of the *bis*-nitro-TAPPs obtained from optical emission and steady-state absorption studies, as well as from DFT computational analysis.**

| TAPP | $\Delta\mu/D^a$ $(\mu_O/D)^b$ | solvent | $\phi_f$ | $\tau/ns^c$ | $k_f \times 10^{-7}/s^{-1\,d}$ $(k_f^{(CT)}; k_f^{(TICT)})^e$ | $k_{nd} \times 10^{-7}/s^{-1\,d}$ $(k_{ISC}^{(CT)}; k_{ISC}^{(TICT)})^e$ |
|---|---|---|---|---|---|---|
| **1p** | 3.9 | cyclohexane | 0.96 | 2.0 | 48 | 2.0 |
| | | CCl$_4$ | 0.93 | 2.5 | 37 | 2.8 |
| | | toluene | 0.70 | 2.0 | 35 | 15 |
| | | 1,4-dioxane | 0.016 | 0.17 | 9.4 | 580 |
| | | THF | 0.030 | 0.14 | 21 | 690 |
| | | ethyl acetate | 0.020 | 0.080 | 25 | 1200 |
| | | SOA | 0.59 | 2.8 | 21 | 18 |
| (DFT) | (1.3) | | | | (53*; 0.44) | (34*; 26) |
| **1m** | 3.8 | cyclohexane | 0.038 | 3.7 | 1.0 | 27 |
| (DFT) | (4.2) | | | | (43; 0.14*) | (84; 24*) |
| **1o** | — | — | — | — | — | — |
| (DFT) | (7.1) | | | | (9.1; 0.0022*) | (5800; 9700*) |
| **2p** | 6.8 | cyclohexane | 1.0 | 1.9 | 53 | <1 |
| | | CCl$_4$ | 0.45 | 2.5 | 18 | 22 |
| | | Toluene | 0.19 | 1.5 | 13 | 54 |
| | | 1,4-dioxane | 0.082 | 0.12 | 68 | 770 |
| | | THF | 0.0023 | 0.96$^f$ | 0.24 | 100 |
| | | ethyl acetate | 0.0020 | 0.79$^f$ | 0.25 | 130 |
| | | SOA | 0.15 | 1.7$^f$ | 8.8 | 50 |
| (DFT) | (1.3) | | | | (73*; 10) | (4.9*; 490) |
| **2m** | 9.0 | cyclohexane | 0.093 | 3.9 | 2.4 | 23 |
| | | CCl$_4$ | 0.0080 | 2.8 | 0.29 | 35 |
| | | toluene | 0.0050 | 1.4 | 0.36 | 71 |
| (DFT) | (9.0) | | | | (72*; 4.2) | (14*; 1200) |
| **2o** | 0.69 | cyclohexane | 0.0006 | 1.1$^f$ | 0.05 | 90 |
| | | toluene | 0.0007 | 2.0$^f$ | 0.04 | 50 |
| | | 1,4-dioxane | 0.0007 | 1.7$^f$ | 0.04 | 60 |
| | | THF | 0.0005 | 2.8$^f$ | 0.02 | 40 |
| (DFT) | (9.0) | | | | (57; 0.0022*) | (51; 3000*) |

$^a$Changes in the electric dipole moments, $\Delta\mu = \mu^* - \mu_0$ (Eq. 1), estimated from the polarity dependence of the Stokes' shifts for non-polar media, i.e., from $d\Delta\tilde{\nu}/d\Delta f_O(\varepsilon, n^2)$ for $\Delta f_O(\varepsilon, n^2) \approx 0$ (Fig. 3c). For **1p**, $r = 8.5$ Å; for **1m**, $r = 8.0$ Å; for **2p**, $r = 12.7$ Å; and for **2m** and **2o**, $r = 11.8$ Å. $^b$Magnitudes of the ground-state permanent electric dipoles calculated from DFT-optimized structures with hexane implemented as a solvation medium. Implementing polar medium, such as DMSO, causes a small increase in the ground-state dipoles, specifically, by 7.7% for **1p** and **2p**, by 14% for **1m**, by 17% for **1o**, by 12% for **2m**, and by 14% for **2o**. $^c$Excited-state lifetimes obtained from deconvolution exponential data fits of emission decays recorded using TCSPC ($\lambda_{ex} = 406$ nm, pulse FWHM = 200 ps). A monoexponential function was implemented for all data fits, except for **2p** and **2o** in some of the solvents, where biexponential fits are required, as described in footnote (f). $^d$Rate constants of radiative and non-radiative decays: $k_f = \phi_f\,\tau^{-1}$, and $k_{nd} = (1 - \phi_f)\,\tau^{-1}$. $^e$Rate constants of radiative decay and ISC obtained from TD DFT calculations for symmetric planer, $S_1^{(CT)}$, and asymmetric twisted, $S_1^{(TICT)}$, excited-state conformers. $^f$Intensity averaged lifetimes, $\bar{\tau} = \Sigma_i \alpha_i \tau_i^2/\Sigma_i \alpha_i \tau_i$, from biexponential data fits: $\tau_i(\alpha_i)$ for **2p** in THF are 0.11 ns (0.995) and 4.9 ns (0.005) yielding $\bar{\tau} = 0.96$ ns and $\langle\tau\rangle = 0.14$ ns, where $\langle\tau\rangle = \Sigma_i \alpha_i \tau_i/\Sigma_i \alpha_i$ is the amplitude average lifetime; for **2p** in ethyl acetate – 0.068 ns (0.98) and 2.0 ns (0.02) yielding $\bar{\tau} = 0.79$ ns and $\langle\tau\rangle = 0.11$ ns; for **2p** in SOA – 0.87 ns (0.66) and 2.3 ns (0.34) yielding $\bar{\tau} = 1.7$ ns and $\langle\tau\rangle = 1.4$ ns; for **2o** in cyclohexane – 0.95 ns (0.98) and 3.0 ns (0.02) yielding $\bar{\tau} = 1.1$ ns and $\langle\tau\rangle = 0.99$ ns; for **2o** in toluene – 0.98 ns (0.78) and 3.1 ns (0.22) yielding $\bar{\tau} = 2.0$ ns and $\langle\tau\rangle = 1.4$ ns; for **2o** in 1,4-dioxane – 1.0 ns (0.97) and 5.5 ns (0.03) yielding $\bar{\tau} = 1.7$ ns and $\langle\tau\rangle = 1.1$ ns; for **2o** in THF – 1.1 ns (0.91) and 5.9 ns (0.09) yielding $\bar{\tau} = 2.8$ ns and $\langle\tau\rangle = 1.5$ ns.

derivatives (Table 1). This finding suggests that the weaker coupling with the *meta* (than *para*) nitro groups, enhances the charge separation in the fluorescent $S_1$ states. The relatively large ground-state dipoles of the *meta* derivatives preclude quadrupole symmetry. The CT in the excited state leads to an increase of the extent of CS. Upon photoexcitation, the dipole of **2m** increases to 18 D, which is equivalent to separating the electron from the hole by about 3.7 Å, This extend of CS is still considerably smaller than the distance between the nitro group and the pyrrolopyrrole. The observed fluorescence suggests that the orbitals, on which the separated charges reside, overlap sufficiently to ensure a considerable $S_1 \longrightarrow S_0$ radiative-transition dipole moment. Furthermore, the values of ($\mu^* - \mu_0$) for **2p** and **2m** are two to three times larger than those for **1p** and **1m** (Table 1). This results is consistent with the larger sizes of the dyes with extra phenylene linkers. Increasing in the donor-acceptor distances increases the separation, $d$, between the charges, $q$, in CT states, and thus increases their permanent dipole moments, i.e., $\mu = q\,d$.

The Stokes' shift of **1o** shows a considerably smaller dependence on solvent polarity than $\Delta\tilde{\nu}$ of the *meta* and *para* derivatives (Fig. 3c). This feature is consistent with the sizable ground-state dipoles of the *ortho bis*-nitro-TAPPs (Table 1), indicating that their ground and the FC excited states have a dipolar character, i.e., the nitro groups in *ortho* position suppress

the formation of symmetric planar structures. In contrary to the accepted notion for A-D-A TAPPs, therefore, there is no quadrupole symmetry breaking in the excited states of **1o** and **2o**. As evident from the obtained values for $\Delta\mu$, photoexcitation of **1o** leads to less than a 10% increase in its dipole (Table 1).

In addition to causing bathochromic shifts in the emission, an increase in solvent polarity quenches the fluorescence of these TAPPs (Fig. 3d, Table 1). It deviates from the behavior reported for some nitro-aromatics that exhibit polarity-induced enhancement of their fluorescence[21–23,61–67]. An increase in solvent polarity drastically reduces the fluorescence quantum yields of **1p**, **2p**, **1m** and **2m** (Table 1). The excited-state lifetimes, estimated from emission decays, follow similar trends (Supplementary Fig. 13). For **1p** and **2p**, the radiative decay rate constants, $k_f$, vary between about $10^8$ and $5 \times 10^8\,s^{-1}$ (Table 1). The radiative-decay rates of **1m** and **2m** are about one-to-two order of magnitude smaller than those of **1p** and **2p**. Variations in solvent polarity induces an order of magnitude changes in $k_f$ of **2m** (Table 1). While solvent effects on $k_f$ are not truly common, they can be prevalent for polarizable and polar media, especially for radiative transitions involving states with a CT character[68–72]. The solvent-induced fluctuations in $k_f$ for **1p** do not exceed a factor of two. For **2p**, an increase in solvent polarity causes about a factor-of-four decrease in $k_f$ before the emission is quenched to the limits of

detection (Table 1). For **2m**, this solvent effect is even more dramatic than for **2p**, causing decrease in $k_f$ (Table 1). These trends show an important characteristic of the *bis*-nitro-TAPPs: decreasing the electronic coupling between the nitro groups and the pyrrolopyrrole core increases the susceptibility of the $S_1 \rightarrow S_0$ radiative transition dipole moment to external perturbations from the solvent media.

The solvent effects on the non-radiative decay constants, $k_{nd}$, are opposite to those on $k_f$ (Table 1). For **1p** and **2p**, increasing solvent polarity induces an increase in $k_{nd}$ that exceeds two orders of magnitude (Table 1). For **2m**, on the other hand, the solvent-induced changes in $k_{nd}$ are moderate (Table 1). The non-radiative decays comprise $S_1 \rightarrow S_0$ (IC), and $S_1 \rightarrow T_n$ (ISC). Medium polarity usually does not affect the rates of ISC. Nevertheless, spin-orbit coupling (SOC) between states with a CT character can lead to some susceptibility of $k_{ISC}$ to solvent polarity[73,74]. Such susceptibility, however, cannot account for the magnitudes of the observed trends with $k_{nd}$ for **1p** and **2p**. Therefore, this drastic polarity-induced increase in $k_{nd}$, leading to the observed fluorescence quenching, originates predominantly from enhanced IC rates. Lowering the energy levels of CT excited states more than the energy of the ground state, brings them closer to the PES of $S_0$ and improves IC. The smaller donor-acceptor coupling in **2m**, indeed, diminishes the CT character of the emissive $S_1$ state and lessens the effects of solvent polarity on IC and $k_{nd}$, which is consistent with the experimental trends (Table 1).

Unlike **1o**, TAPP **2o** is weakly fluorescent with $\phi_f$ between $10^{-4}$ and $10^{-3}$ for different solvents (Figs. 2f and 3d, Table 1). The solvent polarity does not quench the emission of **2o**. While $\phi_f$ shows invariance to solvent polarity, transferring **2o** from non-polar (e.g., cyclohexane) to polar (e.g., DMSO) solvents decreases the lifetime, $\tau$, by a factor of 2.5 (Fig. 3d and Table 1). The nanosecond magnitudes of the measured values for $\tau$ indicate for radiative decays that are more than three orders of magnitude smaller than the non-radiative deactivation rates, i.e., while $k_f$ is between about $2 \times 10^5$ and $5 \times 10^5 \text{ s}^{-1}$, $k_{nd}$ ranges between $4 \times 10^8$ and $9 \times 10^8 \text{ s}^{-1}$ (Table 1). These trends are consistent with the negligible differences between the electric dipole moments (and the polarities) of the ground and the emissive excited state of **2o**, as revealed by the Lippert–Mataga–Ooshika analysis (Fig. 3c and Table 1). The negligible difference between the dipoles of the ground and excited states indicates that they have similar susceptibility to medium polarity. It suppresses solvatochromism and keeps the Stokes' shifts practically the same.

These findings demonstrate the utmost importance of intramolecular CT processes for understanding how to make nitro-aromatics fluoresce. The nitro group provides a high density of close-to isoenergetic triplet and singlet excited states with different symmetries, offering multiple paths for electron transitions between orbitals with different momenta, such as n- and π-bonding, that allows spin flip and ISC, i.e., a classical El Sayed mechanism[75–77]. As an intricate part of the excited-state dynamics of nitro-aromatics, however, CT can dominate the deactivation kinetics producing not only fluorescence enhancement, but also additional pathways for efficient non-radiative deactivation. The emission properties of the *bis*-nitro-TAPPs (Fig. 1) illustrate the balance between these seemingly opposing effects of the nitro groups (Figs. 2, 3).

**Transient-absorption dynamics**. Transient absorption (TA) spectroscopy reveals further details about the excited-state dynamics responsible for the observed photophysics of the *bis*-nitro-TAPPs. TA analysis is especially important for **1o**, the lack of fluorescence for which limits the access to direct observation of the dynamics of its photoinduced behaviour. The TA features of

**1p**, **1m** and **1o** reveals how the position of the nitro groups affects the excited-state dynamics. The outcomes for **1p** and **2p** show the effects of the extra phenylenes. To examine solvent effects, the TA analysis compares the photoinduced kinetics for non-polar environment, i.e., cyclohexane, with that for moderately polar and polar media, i.e., DCM and DMSO.

For cyclohexane, the spectra of **1p** show an overlap of negative $\Delta A$ from the ground-state bleach (B) and the stimulated emission (SE) at $\lambda < 520$ nm with a TA band at 565 nm, which we associate with an $S_1$ transient (Fig. 4a, Table 2). These features for **1p** in non-polar solvent closely resemble the reported TA spectra of pyrrolopyrroles with other electron-withdrawing substituents, in polar media[78]. Global-fit analysis reveals that after a picosecond relaxation, resulting in a slight shifts and changes in the widths of the bands of the TA band (Fig. 5a and Table 2), the $S_1$ state undergoes a multiexponential decay with intensity-averaged lifetime, $\bar{\tau} = 1.9$ ns, which is also obtained from time-resolved fluorescence (Tables 1 and 2). The decay of the SE and the $S_1$ transient accompanies the formation of long-lived weak TA bands, which we ascribe to $T_1 \rightarrow T_{j>1}$ optical transitions (Fig. 4a). Considering the practically quantitative $\phi_f$ of **1p** in cyclohexane (Table 1) suggests that the observed long-lived TA features account for triplet formation with quantum yields that do not exceed a few percent.

An increase in solvent polarity completely alters the TA spectra of **1p** (Fig. 4a vs. b). For DCM and DMSO, a broad TA band at about 550 nm overlaps with SE that extends beyond 700 nm. The SE signal recedes in about a picosecond to further broaden the TA band and extend it into the NIR spectral region. Subsequently, it takes about 10 ps for the broad TA band to decay to baseline (Fig. 4b). These TA features indicate that the deactivation pathways of the photoexcited **1p** in polar media involve multiple excited states, and at least one of them is fluorescent. Observing negative $\Delta A$ in a spectral region where the sample should fluoresce, but does not absorb, is an indication for an emissive excited state. The lack of SE, on the other hand, does not necessarily render a state dark: overlapping with strong TA can make SE undetectable.

Global fits reveal three sequential transitions responsible for the deactivation of the photoexcited **1p** in polar media (Fig. 5b, Table 2): (1) sub-picosecond broadening of the TA peak at about 540 nm with concurrent bathochromic shift of the SE signal, which we observe for DCM; (2) a decrease in SE, accompanying a slight TA decay of the 540-nm peak and appearance of a broad shoulder at $\lambda > 600$ nm, with a rate constant of $1.0 \times 10^{12} \text{ s}^{-1}$ and $8.3 \times 10^{11} \text{ s}^{-1}$ for DCM and DMSO, respectively; and (3) a decay of the broad transient, with rate constants of $1.4 \times 10^{11} \text{ s}^{-1}$ and $9.1 \times 10^{10}$ for DCM and DMSO, respectively.

Global fits employing models with multiple parallel pathways, i.e., $S_1^{(I)} \longrightarrow X_1^{(II)} + X_1^{(III)} + S_0$, $X_1^{(II)} \longrightarrow X_1^{(III)} + S_0$ and $X_1^{(III)} \longrightarrow S_0$, show negligible to no contribution from direct decays of $S_1^{(I)}$ and $S_1^{(II)}$ to the ground state and from a direct transition from $S_1^{(I)}$ to $S_1^{(III)}$. For example, for branching pathways, characterized with rate constants $k_1 = \tau_1^{-1}$ and $k_2 = \tau_2^{-1}$, when $k_1 >> k_2$, the kinetic analyses do not reliably discern between $\alpha(\lambda) \exp(-(\tau_1^{-1} + \tau_2^{-1}) t)$ and $\alpha(\lambda)\exp(-\tau_1^{-1} t)$ when the involved transients have similar extinction coefficients at $\lambda$, leaving the pathway characterized with $\tau_2$ undetectable (Supplementary Table 2). Therefore, the sequential transitions, $S_1^{(I)} \longrightarrow X_1^{(II)} \longrightarrow X_1^{(III)} \longrightarrow S_0$, represent the principal kinetic contribution to the observed TA dynamics for **1p** in polar media. With X = S or T, the model does not claim a prior knowledge of the multiplicity of the involved excited states. Nevertheless, the lifetime of **1p** triplet for non-polar media extends considerably beyond the nanosecond dynamic range of the pump-probe technique. It is quite unlikely that an increase in solvent polarity increases the

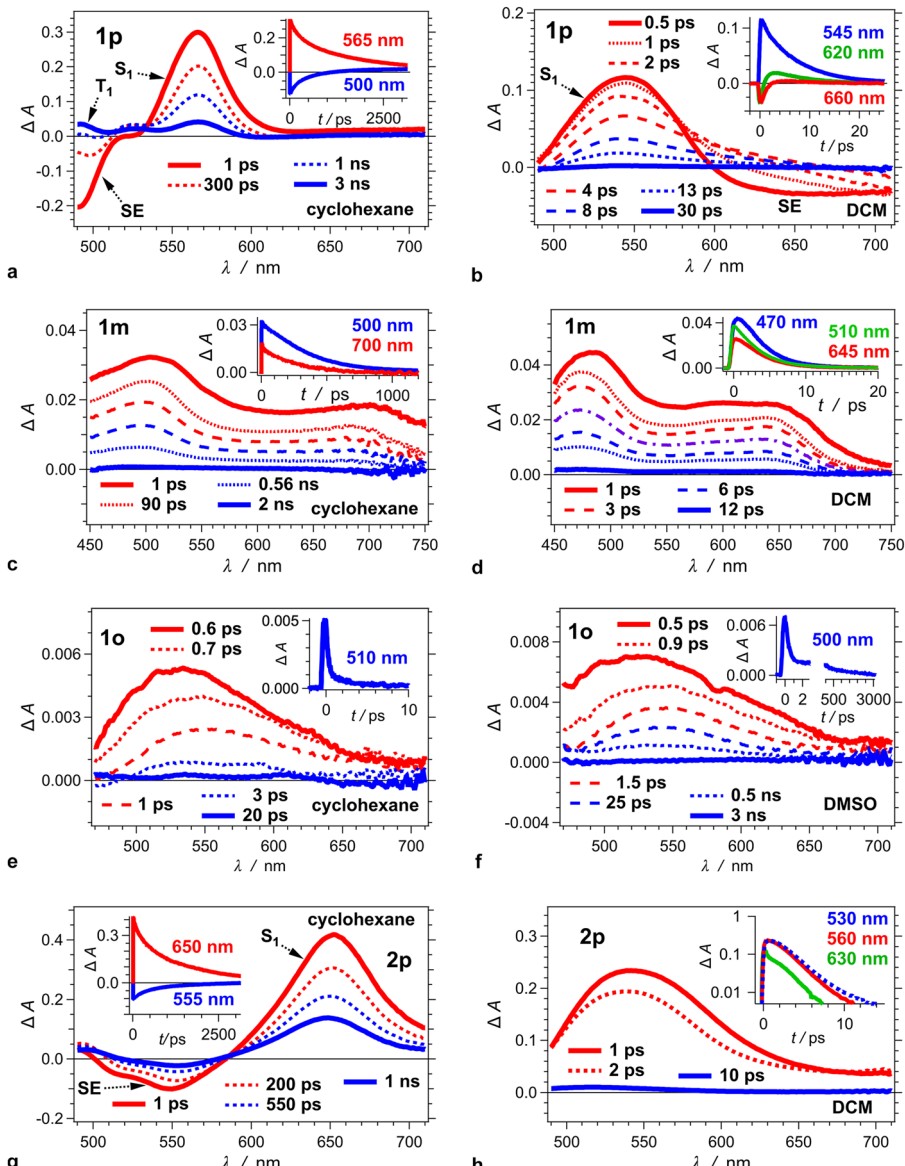

**Fig. 4 Transient absorption (TA) dynamics of *bis*-nitro-TAPPs in organic solvent media with various polarities. a, b, g, h** TA for **1p** and **2p** showing absorption features of the $S_1$ and $T_1$ transients, along with the stimulated emission (SE). **c, d** TA for **1m** showing features of $S_1$ transient comprising a band at about 500 nm, ascribed to the absorption of the radical cation of the nitrophenyl acceptors, and a broadband extending to 750 nm, ascribed to the absorption of the radical cation of pyrrolopyrrole. **e, f** TA of **1o** showing a board featureless band that decays in two discussant timescales: a principal subpicosecond decay, followed by deactivation in the picosecond (for nonpolar solvents) and nanosecond (for polar solvents) time domains. ($\lambda_{ex} = 400$ nm; 4 μJ per pulse at a repetition rate of 1 kHz, and the pump blocked every other pulse; pulse FWHM = 50 fs; data recorded in a transmission mode through a 2-mm quartz cuvette, which increases the effective instrument response time to about 200 fs). The graphs represent the TA spectra recorded at different times; and the insets show the TA kinetic curves recorded at different wavelengths.

rates of the $T_1 \longrightarrow S_0$ transition by several orders of magnitude. Hence, the excited-state picosecond dynamics of **1p** for polar-media appears confined to the singlet manifold, i.e., X = S. Furthermore, the lack of long-lived transients of **1p** in DCM and DMSO suggests that efficient IC dominates the non-radiative deactivation of $S_1$ to $S_0$ state, i.e., the state with 10-ps lifetime is a minimum on the $S_1$ PES that forms a CI with $S_0$.

These solvent effects on TA dynamics of **1p** are consistent with the propensity of such A–D–A dyes for ES-SB[56]. For non-polar media, the FC $S_1$ state undergoes a picosecond relaxation into a structure with similar nuclear and electronic configurations, and a slightly increased CT character[56,79]. An increase in solvent polarity not only accelerates this CT process, but also leads to CT states with improved charge separation[56,79].

Moving the nitro groups to the *ortho* and *meta* positions drastically changes the TA spectra. While an increase in solvent polarity accelerates the deactivation of the excited states of **1m** and **1o**, it does not perturb much the appearance of their TA spectra (Fig. 4c–f).

In non-polar media, **1m** behaves like **1p** in polar solvents, manifesting a picosecond decay with 10-to-30-nm hypsochromic shifts without forming long-lived transients (Fig. 4c, Table 2). An increase in solvent polarity shortens the lifetimes of the excited states of **1m** by one to two orders of magnitude (Table 2). As unveiled by global fits for the TA of **1m**, sequential transformations, involving two excited-state transients with similar spectral features, can readily account for the observed excited-state dynamics (Fig. 5c, Table 2). The fast picosecond rates of decay

**Table 2 Summary of the TA features of 1p, 1m, 1o, and 2p for cyclohexane, DCM, and DMSO.**

| | cyclohexane | DCM | DMSO |
|---|---|---|---|
| **1p** | **spectral peaks:** | | |
| | $S_1^{(I)}$: 566 nm (A), <520 nm (SE, B) | $S_1^{(I)}$: 538 nm (A), 614 nm (SE) | $S_1^{(II)}$: 553 nm (A), 685 nm (SE) |
| | $S_1^{(II)}$: $S_1^{(III)}$: 567 nm (A), <520 nm (SE, B) | $S_1^{(II)}$: 542 nm (A), 680 nm (SE) | $S_1^{(III)}$: 552 nm (A) |
| | $T_1$: 494 nm (A), 528 nm (A) | $S_1^{(III)}$: 544 nm (A) | |
| | **kinetics:** | | |
| | $S_1^{(I)} \longrightarrow S_1^{(II)} + S_1^{(III)} \longrightarrow S_0 + T_n$ | $S_1^{(I)} \longrightarrow S_1^{(II)} \longrightarrow S_1^{(III)} \longrightarrow S_0$ | $S_1^{(II)} \longrightarrow S_1^{(III)} \longrightarrow S_0$ |
| | $S_1^{(I)} \longrightarrow S_1^{(II)} + S_1^{(III)}$: 3.3 ps | $S_1^{(I)} \longrightarrow S_1^{(II)}$: 0.2 ps | $S_1^{(II)} \longrightarrow S_1^{(III)}$: 1.2 ps |
| | $S_1^{(II)} + S_1^{(III)} \longrightarrow S_0 + T_n$: | $S_1^{(II)} \longrightarrow S_1^{(III)}$: 1.0 ps | $S_1^{(III)} \longrightarrow S_0$: 11 ps |
| | 180 ps (0.30); 2.0 ns (0.70) | $S_1^{(III)} \longrightarrow S_0$: 7.3 ps | |
| **1m** | **spectral peaks:** | | |
| | $S_1^{(I)}$: 514 nm (A), 705 nm (A) | $S_1^{(I)}$: 500 nm (A), 673 nm (A) | $S_1^{(I)}$: 482 nm (A), 670 nm (A) |
| | $S_1^{(II)}$: 480 nm (A); 530 & 710 nm (SE) $S_1^{(III)}$: 504 nm (A), 690 nm (A) | $S_1^{(II)}$: 470 nm (A), 642 nm (A) | $S_1^{(II)}$: 466 nm (A), 644 nm (A) |
| | **kinetics:** | | |
| | $S_1^{(I)} \longrightarrow S_1^{(II)} + S_1^{(III)}$; $S_1^{(II)} \longrightarrow$; $S_1^{(III)} \longrightarrow S_0$ | $S_1^{(I)} \longrightarrow S_1^{(III)} \longrightarrow S_0$ | $S_1^{(I)} \longrightarrow S_1^{(III)} \longrightarrow S_0$ |
| | $S_1^{(I)} \longrightarrow S_1^{(II)}$: 160 ps | $S_1^{(I)} \longrightarrow S_1^{(III)}$: 1.0 ps | $S_1^{(I)} \longrightarrow S_1^{(III)}$: 0.60 ps |
| | $S_1^{(I)} \longrightarrow S_1^{(III)}$: 38 ps | $S_1^{(III)} \longrightarrow S_0$: 3.9 ps | $S_1^{(III)} \longrightarrow S_0$: 3.4 ps |
| | $S_1^{(II)} \longrightarrow S_0$: 300 ps | | |
| | $S_1^{(III)} \longrightarrow S_0$: 3.7 ns | | |
| **1o** | **spectral peaks:** | | |
| | $S_1^{(I)}$: 532 nm (A) | $S_1^{(I)}$: 547 nm (A) | $S_1^{(I)}$: 525 nm (A) |
| | $S_1^{(II)}$: 552 nm (A) | $S_1^{(II)}$: 577 nm (A) | $S_1^{(II)}$: 543 nm (A) |
| | | | X: 540 nm (A) |
| | **kinetics:** | | |
| | $S_1^{(I)} \longrightarrow S_1^{(II)} \longrightarrow S_0$ | $S_1^{(I)} \longrightarrow S_1^{(II)} \longrightarrow S_0$ | $S_1^{(I)} \longrightarrow S_1^{(II)} \longrightarrow X + S_0$; $X \longrightarrow S_0$ |
| | $S_1^{(I)} \longrightarrow S_1^{(II)}$: 0.35 ps | $S_1^{(I)} \longrightarrow S_1^{(II)}$: 0.44 ps | $S_1^{(I)} \longrightarrow S_1^{(II)}$: 0.55 ps |
| | $S_1^{(II)} \longrightarrow S_0$: 5.8 ps | $S_1^{(II)} \longrightarrow S_0$: 13 ps | $S_1^{(II)} \longrightarrow S_0$ :380 ps |
| | | | $S_1^{(II)} \longrightarrow X$: 44 ps |
| | | | $X \longrightarrow S_0$: 1.3 ns |
| | | | (X = $S_1^{(III)}$ or T) |
| **2p** | **spectral peaks:** | | |
| | $S_1^{(I)}$: 651 nm (A), 543 nm (SE) | $S_1^{(I)}$: 591 (A) | $S_1^{(I)}$: 597 (A) |
| | $S_1^{(II)}$: 658 nm (A), 516 & 551 nm (SE) | $S_1^{(II)}$: 535 (A) | $S_1^{(II)}$: 544 (A) |
| | $S_1^{(III)}$: 649 nm (A), 553 nm (SE) | | |
| | **kinetics:** | | |
| | $S_1^{(I)} \longrightarrow S_1^{(II)} + S_1^{(III)} \longrightarrow S_0$ | $S_1^{(I)} \longrightarrow S_1^{(II)} \longrightarrow S_0$ | $S_1^{(I)} \longrightarrow S_1^{(II)} \longrightarrow S_0$ |
| | $S_1^{(I)} \longrightarrow S_1^{(II)} + S_1^{(III)}$: 1.8 ps | $S_1^{(I)} \longrightarrow S_1^{(II)}$: 0.68 ps | $S_1^{(I)} \longrightarrow S_1^{(II)}$: 0.41 ps |
| | $S_1^{(II)} + S_1^{(III)} \longrightarrow S_0$: | $S_1^{(II)} \longrightarrow S_0$: 2.4 ps | $S_1^{(II)} \longrightarrow S_0$: 2.7 ps |
| | 220 ps (0.39); 1.9 ns (0.61) | | |

to the ground state for polar media, renders the ascription of singlet character of these excited states feasible. The longest lifetime for **1m** in cyclohexane fits with functions describing such sequential kinetics, however, is it shorter than 400 ps, which does not account for the 3.7-ns decay obtained from time-resolved emission analysis (Table 1). Introducing this nanosecond lifetime as a fixed parameter to the global fits leads to the emergence of an alternative kinetic pattern involving two parallel pathways (Table 2). One of them involves sequential transitions similar to those observed for polar media, with a rate constant of $2.6 \times 10^{10}$ s$^{-1}$ for the first step followed by a decay to $S_0$ with $3.3 \times 10^9$ s$^{-1}$ (Table 2). The other, originating from the same initial excited state, involves considerably slower sequential steps with rate constants $6.3 \times 10^9$ s$^{-1}$ and $2.7 \times 10^9$ s$^{-1}$. The TA spectra of the two short-lived transients comprise a peak at about 500 nm and a broadband extending to the red and NIR regions. The spectrum of the long-lived transient, on the other hand, shows features consistent with overlapping TA and SE bands resulting in $\Delta A$ amplitudes that are an order of magnitude smaller than those of the short-lived states. The quantum yield of forming the long-lived transient is about 0.2, which along with the weak spectral features, makes its $\Delta A$ peak amplitudes at 2 ns about 100 smaller than the initial TA amplitudes of **1m**. As a results, observing the TA of the fluorescent state of **1m** in the nanosecond time domain is implausibly challenging, while its emission is readily measured.

Therefore, the complementarity between emission and TA spectroscopy (Tables 1 and 2) is key for gaining understanding of the excited-state dynamics.

The TA spectra of **1m** show a peak between 450 and 550 nm, along with a broadband that extends to 700 nm (Figs. 4c, d and 5c). The broadband is characteristic of the absorption of oxidized pyrrolopyrroles[80], while the peak at about 500 nm can be ascribed to the radical cation of nitrobenzene[81]. That is, regardless of the medium polarity, photoexcitation of **1m** yields predominantly a CT state with well-separated charges that do not contribute to the detected fluorescence.

For all solvents, a broadband dominates the TA spectra of **1o**, which resembles the TA of **1p** in polar media after the SE decays away (Fig. 4e, f). Unlike **1m**, however, for **1o** relatively slow decays follow the initial sub-picosecond deactivation (Fig. 4e, f). These transitions of **1o** involve transients with quite similar absorption features: i.e., a broadband at 550 nm with a shoulder that into the red spectral region.

Adding electron-donating substituents to the *ortho* and *para* positions of nitrobenzene broadens and bathochromically shifts the absorption of its radical anion[81,82]. The TA spectra for **1m** are consistent with an A-D$^{\bullet+}$-A$^{\bullet-}$ CT state with distinct bands for D$^{\bullet+}$ and A$^{\bullet-}$ (Fig. 4c, d). For **1p** (Fig. 4b) and **1o** (Fig. 4e, f), on the other hand, the improved donor-acceptor electronic coupling broadens the absorption of A$^{\bullet-}$, causing an overlap with that of

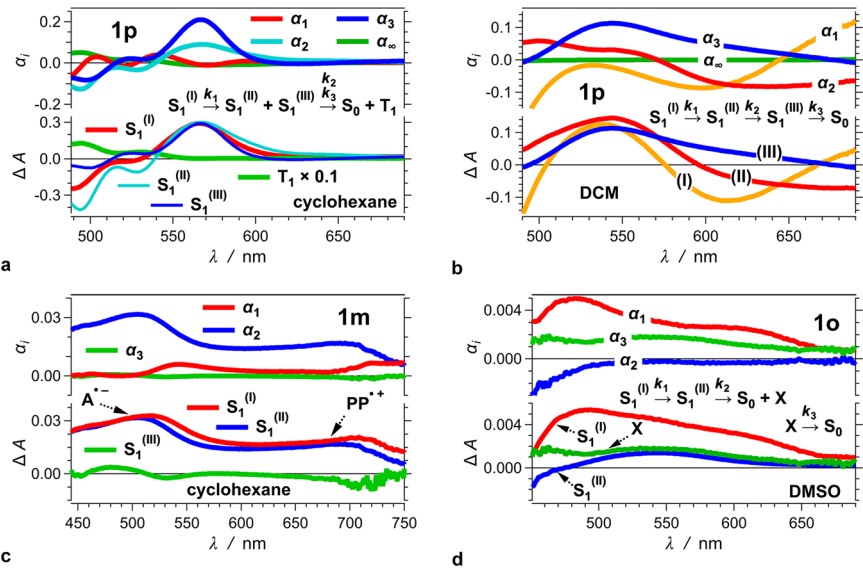

**Fig. 5 Global-fit analysis of transient absorption spectra and kinetics of 1p, 1m, and 1o.** Global-fit analysis of transient absorption (TA) spectra and kinetics of **1p**, **1m**, and **1o**, using multiexponential functions, i.e., $\Delta A(\lambda, t) = \alpha_\infty(\lambda) + \Sigma_i \alpha_i(\lambda) \exp(- t/\tau_i)$; $k_i = \tau_i^{-1}$; and for parallel transitions with rate constants $k_i^{(1)}$ and $k_i^{(2)}$, $k_i = k_i^{(1)} + k_i^{(2)}$. A positive (negative) amplitude, $\alpha_i(\lambda)$, corresponds to a decay (rise) of TA, or depletion (growth) of SE or ground-state bleach at wavelength $\lambda$. In each panel, the upper graphs represent the amplitude spectra, $\alpha_i(\lambda)$, of each transition, and the lower—the extracted of the different states. The numeration of the $S_1$ states with (I), (II), and (III) is arbitrary and does not necessarily invoke a direct relationship with the $S_1^{(FC)}$, $S_1^{(CT)}$, and $S_1^{(TICT)}$ structures obtained from TD DFT computational analysis. **a** All singlet-excited-states of **1p** in cyclohexane have quite similar TA spectra ($\tau_1 = 3.3$ ps, $\tau_2 = 180$ ps, $\tau_3 = 2.0$ ns, the value of $\tau_3$ is from TCSPC analysis, Table 1). After 3-ps spectral shift, depicted by $\alpha_1(\lambda)$ and $\tau_1$ and corresponding to the formation of $S_1^{(II)}$ and $S_1^{(III)}$ from $S_1^{(I)}$. $\alpha_2(\lambda)$ and $\tau_2$ depict the decay of $S_1^{(II)}$, and $\alpha_3(\lambda)$ and $\tau_3$ - of $S_1^{(III)}$. Effectively, $\tau_2$ and $\tau_3$ represent a biexponential decay to $S_0$. As the fluorescence analysis of **1p** in cyclohexane suggests (Table 1), the rate of triplet formation cannot exceed $2 \times 10^7$ s$^{-1}$ which is outside the dynamic range of the pump-probe technique. Concurrently, the quantum yield of triplet formation cannot exceed 0.04, which suggests that the triplet spectral features are considerably more intense than $\alpha_\infty(\lambda)$. **b** **1p** in DCM shows sequential transformations between picosecond-lived singlet excited states ($\tau_1 = 0.2$ ps, $\tau_2 = 1.0$ ps, $\tau_3 = 7.3$ ps). When introduced to the fitting function, parallel pathways, such as $S_1^{(I)} \longrightarrow S_0$, $S_1^{(I)} \longrightarrow S_1^{(II)}$, and $S_1^{(II)} \longrightarrow S_0$, show negligible to no contribution to the TA kinetics. **c** The $S_1^{(I)}$ and $S_1^{(II)}$ stats of **1m** show pronounced charge separation, as evident from the spectral features of the oxidized nitrophenyl acceptors, A$^{•-}$, and the reduced pyrrolopyrrole donor, PP$^{•+}$ [82,83]. ($\tau_1 = 31$ ps, $\tau_1^{-1} = \tau_{11}^{-1} + \tau_{12}^{-1}$ represents initiation of parallel pathways, $\tau_{11} = 160$ ps, $\tau_{12} = 38$ ps, $\tau_2 = 300$ ps, $\tau_3 = 3.7$ ns, the value of $\tau_3$ is from TCSPC analysis, Table 1). Considering the substantial ground-state dipole of **1m**, i.e., exceeding 4 D (Table 1), it is safe to assume that the FC excited state has a CT character, and the equally large $\Delta\mu$ suggests that relaxation along the $S_1$ PES increases the extent of CS. **d** Dominated by sub-picosecond deactivation, the TA features of **1o** show transitions between transients with strongly overlapping spectral features ($\tau_1 = 0.55$ ps, $\tau_2 = 39$ ps, $\tau_2^{-1} = \tau_{21}^{-1} + \tau_{22}^{-1}$ represents initiation of parallel pathways, $\tau_{21} = 380$ ps, $\tau_{22} = 44$ ps, $\tau_3 = 1.3$ ns).

D$^{•+}$, which brings subtilty to the ascription of a CT character to the observed transitions. Nevertheless, a spectral deconvolution shows the contribution of two overlapping bands in the region between 500 and 700 nm, which are consistent with assignments to the oxidized pyrrolopyrrole and the reduced nitrophenyls composing excited states with a CT character (Figs. 4c, d and 5c).

The broad featureless band in the TA spectra of **1o** undergoes a sub-picosecond decay with $\tau$ between 300 and 600 fs, accompanied by a bathochromic shift (Fig. 4e, f). Ascribing this short-lived transient to an $S_1$ state suggests that it undergoes a radiative decay with $k_f < 3 \times 10^8$ s$^{-1}$, considering undetectable fluorescence, i.e., $\phi_f \lesssim 10^{-4}$.

Global-fit analysis reveals that the sub-picosecond deactivation of the photoexcited **1o** leads to another excited-state transient, which decays in about 5 to 30 ps in different solvents (Table 2). The TA decay of **1o** shows dependence on medium polarity (Table 2). For cyclohexane and DCM, the slow decay of **1o** proceeds with rate constants of $1.7 \times 10^{11}$ s$^{-1}$ and $7.7 \times 10^{10}$ s$^{-1}$, respectively (Table 2). The TA analysis for **1o** in DMSO, however, presents a different paradigm (Fig. 5d). The sub-picosecond decay leads to a transient, $S_1^{(II)}$, with a lifetime of about 25 ps. While this transient shows a decay to the ground state with a rate constant of about $2.6 \times 10^9$ s$^{-1}$, it predominantly leads to the formation of a third transient, X, with about a ten times faster rate, $\tau^{-1} = 2.3 \times 10^{10}$ s$^{-1}$ (Table 2). The lifetime of this transient,

X, extends beyond 1 ns, which is well within the detection range of time-correlated single-photon counting (TCSPC) used for recording the emission decays (Table 1). The lack of detectable fluorescence for **1o** suggests that this long-lived species correspond to a dark excited state, i.e., with $k_f < 10^5$ s$^{-1}$.

The model does not provide information about the multiplicity of the involved excited states. Nevertheless, we ascribe a triplet character to X assuming $k_{ISC}$ of about $2 \times 10^{10}$ s$^{-1}$. In 30 ps, more than 99% of $S_1^{(II)}$ decays to $S_0$ for cyclohexane, and more than 90% for DCM. For these solvents, therefore, 40- or 50-ps ISC does not lead to significant amounts of triplets that are necessarily reliably detectable. With considerably slower decays to ground state in DMSO, about 90% of $S_1^{(II)}$ still remains after 50 ps. This solvent dependence of the rates of $S_1^{(II)} \longrightarrow S_0$ makes ISC a probable pathway for non-radiative deactivation of **1o** in DMSO. Computational analysis yielding $k_{ISC} > 10^{10}$ s$^{-1}$ for **1o**, which is more than two orders of magnitude larger than the calculated $k_{ISC}$ for **1p** and **1m** (Table 1), is consistent with ascribing a triplet character to the long-lived state, X.

Moving the nitro groups to the *ortho* position appears to dramatically increase the rates of ISC. For **1p** in cyclohexane, the non-radiative decay rate constant is in the order of $10^7$ s$^{-1}$ (Table 1), and $k_{ISC} \leq k_{nd}$ that makes the triplet TA barely noticeable. For **1m**, TA does not reveal patterns that are ascribable to triplet formation even when the excited-state

lifetime is 360 ps. For **1o** in DMSO, ISC appears to occur at time scales shorter than 100 ps. In addition to accelerating triplet formation, i.e., $S_1 \longrightarrow T_j$, placing the nitro groups at *ortho* position also increases the ISC rates of triplet deactivation, i.e., $T_1 \longrightarrow S_0$. The lifetime of the longest living transient of **1o** barely exceeds 1 ns (Fig. 4f, Table 2). Triplet lifetimes, $^3\tau$, of organic chromophores usually extend to hundreds of microseconds. Nevertheless, vibrational modes of nitro groups in triplet states with a pronounced CT character can significantly shorten $^3\tau$ and push it into the sub-nanosecond time domain[76,83,84].

Introducing phenylene linkers to **1p** does not change the trends in the TA spectra, but shortens the lifetimes of the excited states by about a factor of two. For cyclohexane, **2p** shows a distinct TA band, with a maximum at 650 nm, along with SE at 550 nm (Fig. 4g). Following the initial 2-ps relaxation, the simultaneous multiexponential decay of the TA band and the decrease of the SE intensity form an isosbestic point (Table 2). This TA dynamics illustrates a deactivation of an emissive state with an intensity-averaged lifetime of 1.8 ns (Table 1).

Similar to **1p**, the TA spectra of **2p** for polar media reveal a broad TA band at about 550 nm and a shoulder extending into the red spectral region that undergo a picosecond decay without forming long-lived transients (Fig. 4h, Supplementary Fig. 14). The 25-nm hypsochromic shift of the peak as TA decays, and the slight nonlinearity in $lg(\Delta A)$ vs. $t$ (Fig. 4h) suggest for kinetics involving multiple excited states. The global-fit analysis reveals a sequential transition in the sub-picoecond and picosecond time domains. These shorter excited-state lifetimes of **2p**, in comparison with **1p**, are consistent with extra conformational degrees of freedom, from the additional phenylenes, increasing the number of possible pathways for non-radiative decay.

The emerging patterns from the TA analysis of the *bis*-nitro-TAPPs reveals that IC is by far the principal mode for deactivation of their singlet excited states. Except for **1p** in cyclohexane and **1o** in DMSO, TA does not reveal convincing evidence for triplet formation. The CT character introduced by the nitro groups in many of the states of these dyes makes their excited-state kinetics susceptible to medium polarity. When an excited state has a stronger CT character than the ground sate, an increase in media polarity will bring their PESs closer together and accelerate the non-radiative decay via IC, which is the case for most of the *bis*-nitro-TAPPs as TA spectroscopy reveals. Conversely, when an excited state has a weaker CT character than the ground sate, the rates of IC decrease with an increase in solvent polarity, which appears to be the case for **1o** (Table 2).

Conformational dynamics is essential for the conjugates in their excited states to explore the PES and converge into the CIs mediating IC to the ground state. An increase in the torsional degrees of freedom not only aids this convergence along the $S_1$ PES, but also can increase the number CIs between $S_1$ and $S_0$. As a result, the extra phenylene linkers of **2p** tend to shorten its excited-state lifetimes, especially for polar media (Table 2, Supplementary Fig. 15).

**Solid solvent medium**. To further examine the effects of molecular flexibility on the fluorescence properties of the *bis*-nitro-TAPPs, we resort to a solid glass as a solvent medium. At temperatures above ~87 °C, sucrose octaacetate (SOA) exists as a viscous liquid. Upon cooling to room temperature, SOA solidifies to form a single-block glassy substance with optical transparency extending over the visible spectral range[85]. Dissolving *bis*-nitro-TAPPs in molten SOA and allowing it to cool down to room temperature yield solutions of these dyes in a glass medium. Solutions of the same TAPPs in another ester solvent, ethyl acetate (EA), serve as representative controls. For this study, we

focus on **1p** and **2p** because they exhibit detectable fluorescence in EA.

When placed in solid media, **1p** and **2p** exhibit a 30-to-70-fold increase in their fluorescence quantum yields (Fig. 2g, h, Table 1). Enhancement of molecular rigidity via solidifying the media and lowering the temperature, indeed, strongly suppresses the non-radiative deactivation pathways[86–88], which leads to an increase in the emission quantum yields, representing both fluorescence and phosphorescence processes[86,89]. At room temperature, while the SOA solid medium suppresses large-amplitude conformational changes, it does not truly affect vibrational transitions that require only a small perturbation in the volume of the solvation cavity. Therefore, the SOA-induced fluorescence enhancement of **1p** and **2p** originates from suppressing the torsional modes of the aromatic rings.

In addition to enhancing $\phi_f$, SOA decreases the Stokes' shifts, $\Delta\tilde{\nu}$, of **1p** and **2p** in comparison with EA (Fig. 2g, h, Table 1). Considering the SOA-induced suppression of structural reorganization, these findings suggest that the molecular geometries of the $S_1$ states that are closer to the FC structures have a higher propensity for fluorescence than the relaxed $S_1$ conformers.

In this analysis, we ascribe the SOA-induced increase in $\phi_f$ and decrease in $\Delta\tilde{\nu}$ to the increase in the rigidity of the solvation cavities of the TAPPs. Solidifying of solvation media, however, inherently lowers its dielectric constant because of suppression of the orientational polarization[90], and polarity may account for the observed difference between the solid SOA and liquid EA samples. While SOA and EA have practically the same relative abundance of the dipolar ester groups, SOA has extra ether oxygens that enhance its polarity. Therefore, ester solvents, such as EA and diethyl sebacate with static dielectric constants between about 5 and 6, serve as excellent liquid controls with similar polarity to SOA solid solutions[85,91]. Furthermore, the values of $\phi_f$ and $\Delta\tilde{\nu}$ for **1p** and **2p** in SOA are quite similar to those for non-polar solvents, such as toluene, $CCl_4$, and 1,4-dioxane, rather than to those for slightly more polar solvents, such as THF and EA. Therefore, the suppression of conformational changes along the PES of $S_1$ represents the principal contribution to the observed SOA-induced fluorescence enhancement and $\Delta\tilde{\nu}$ of the *bis*-nitro-TAPSs.

**Computational analysis**. To gain further insight into how the fluorescent properties of the *bis*-nitro-TAPPs originate from the balance between the rates of radiative decay and the competing non-radiative deactivation involving ISC and IC, we resort to TD-DFT and unrestricted DFT calculations at a CAM-B3LYP/6-31 +G(d) level of theory. COSMO continuum solvation model allows us to implement hexane as a solvent for these calculations[92].

The free rotation around the carbon–carbon single σ-bonds with the nitrophenyl electron acceptors have deterministic effects on the excited-state dynamics of these TAPPs. The computational analysis reveals that such dihedral exploration of the $S_1$ PES allows the Franck-Condon state, $S_1^{(FC)}$, formed upon photo-excitation, $S_0 \rightarrow S_1^{(FC)}$, to relax into two principal minima with a CT character: one, $S_1^{(CT)}$, structurally similar to the $S_1^{(FC)}$ state, and another with the characteristics of a twisted intramolecular charge-transfer (TICT) state, $S_1^{(TICT)}$ (Fig. 6). Furthermore, the $S_1^{(CT)}$ conformers closely resemble the structures of the lowest triplet state, $T_1$. The energy levels of the twisted triplets, $T_1^{(t)}$, which have the $S_1^{(TICT)}$ geometries, are always above those of $T_1$ (Fig. 6, Supplementary Figs. 11, 12).

The calculations reveal that the $S_1^{(CT)}$ states can undergo radiative deactivation, $S_1^{(CT)} \rightarrow S_0^{(CT,FC)}$, with oscillator strength, $f$, for the different *bis*-nitro-TAPPs ranging between 0.5 and 2.7.

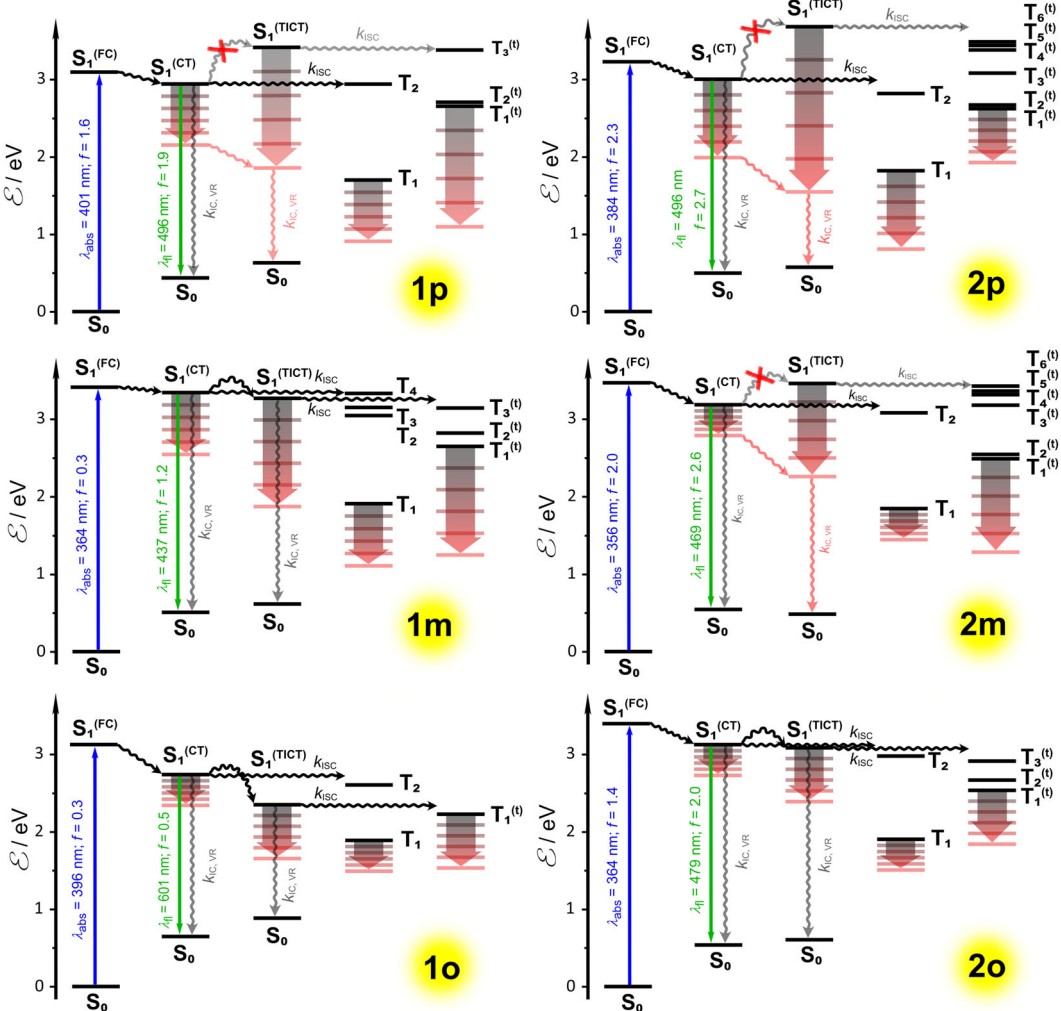

**Fig. 6 Simplified Jablonski diagrams for all of the studied molecules 1p,m,o and 2p,m,o.** Constructed on the basis of B3LYP/6-31 G(d) optimized geometries of $S_0$, $S_1^{(IC)}$, and $S_1^{(TICT)}$ states and TDDFT/cam-B3LYP/DZP calculations of vertical transitions energies. The gradients from grey to red indicate an increase in the solvent polarity.

The $S_1^{(TICT)}$ states, on the other hand, are dark with $f$ for $S_1^{(TICT)} \rightarrow S_0^{(TICT,FC)}$ not exceeding 0.26 (**2p**). Thus, non-radiative processes, i.e., ISC and IC, represent the principal pathways of deactivation of the $S_0^{(TICT,FC)}$ states. Initial evaluations show that TAPPs, such as **1p**, with an energy level of $S_1^{(TICT)}$ above that of $S_1^{(CT)}$, are fluorescent; and when $S_1^{(TICT)}$ is below $S_1^{(CT)}$ like in the case of **1o**, the TAPPs do not exhibit detectable emission (Figs. 2 and 6).

In the optimized geometries of the ground states of the *bis*-nitro-TAPPs, the -$C_6H_4NO_2$ acceptors assume dihedral angles with the pyrrolopyrrole donor (for **1o**, **1m**, and **1p**) and with the -$C_6H_4$- linkers (for **2o**, **2m**, and **2p**) that range between 30 and 40 degrees for **1p**, **2p**, **1m** and **2m** and about 50 degrees for **1o** and **2o** where the steric hindrance from the *ortho* nitro groups has a noticeable effect (Fig. 1, Supplementary Fig. 10 and Supplementary Table 3). With both dihedral angles the same, ranging between about 20 and 40 degrees, the $S_1^{(CT)}$ states of the *bis*-nitro-TAPPs exhibit an increase in planarity in comparison with the $S_0$ and $S_0^{(FC)}$ structures. An increase in the orthogonality between the -$C_6H_4NO_2$ moieties and the aromatic rings to which they are connected, leads to the asymmetric $S_1^{(TICT)}$ states with improved charge-separation (CS) character. That is, while the overlap between the transition orbitals for the $S_1^{(CT)}$ conformers ensures efficient radiative $S_1^{(CT)} \rightarrow S_0^{(CT,FC)}$ deactivation, the lack

of such a good orbital overlap in the TICT structures makes the $S_1^{(TICT)}$ states dark (Supplementary Fig. 10).

Interestingly, the computed rates of radiative deactivation, $k_f$, correlate well with the decrease in the dihedral angles with the -$C_6H_4NO_2$ moieties. For the $S_1^{(CT)}$ states, $k_f$ assumes $5.3 \times 10^8$ s$^{-1}$ for **1p**, $4.3 \times 10^8$ s$^{-1}$ for **1m**, and $9.1 \times 10^7$ s$^{-1}$ for **1o**. These computed rate constants agree well with the experimentally determined $k_f$ values for **1p**, but appear as overestimates for **1m** (Table 1).

Twisting of the -$C_6H_4NO_2$ ring out of plane leads to a drastic decrease in $k_f$, i.e., $4.4 \times 10^6$ s$^{-1}$ for **1p**, $1.4 \times 10^6$ s$^{-1}$ for **1m**, and $2.2 \times 10^4$ s$^{-1}$ for **1o** represent the $S_1^{(TICT)} \rightarrow S_0^{(TICT,FC)}$ radiative transitions. The computed $k_f$ values for the $S_1^{(CT)}$ and $S_1^{(TICT)}$ conformers of the TAPPs with the extra phenylene linkers, **2p**, **2m**, and **2o**, exhibit similar trends. The decrease in $k_f$ with moving the nitro groups from *para* to *ortho* positions accompanies the lowering of the energy level of the $S_1^{(TICT)}$ below that of $S_1^{(CT)}$ (Fig. 6). Both of these effects synergistically enhance the decrease in $\phi_f$, which is consistent with the experimental observations.

The trends in the radiative dynamics and the conformer energies in the singlet manifolds illustrate well the underlying reasons for the observed photophysics of the *bis*-nitro-TAPPs. Nevertheless, the high propensity of nitro-aromatics for triplet

formation warrants examination of the SOC with the corresponding triplet states, along with the rates of ISC from $S_1^{(CT)}$ and $S_1^{(TICT)}$ to the $T_j$ and $T_j^{(t)}$ states, respectively, with identical geometries and equal or lower energy levels. The transitions from $S_1$ to the relatively planar lowest triplet excited states, $T_1$, have negligibly low probability. Therefore, the ISC transitions to the upper triplet states, i.e., to $T_{(2 \text{ to } j)}$ and $T_{(1 \text{ to } j)}^{(t)}$, define $k_{ISC}$. Because, the $T_j$ and $S_1^{(CT)}$ states have the same orbital symmetry, the corresponding SOC matrix elements are very small, i.e., less than $1 \text{ cm}^{-1}$, except $\langle S_1 | \hat{H}_{SO} | T_2 \rangle$ for the $\textbf{1o}$. It means that the energy gap is the main factor that affects the ISC rates: i.e., the smaller $\Delta\mathscr{E}(S_1-T_j)$, the bigger FC factor ($F_{0m}$) is. As a result, transitions between the closest lying $S_1$ and $T_j$ states dominantly contribute to the total ISC rates. At the same time, the large energy gap between $S_1^{(CT)}$ and $T_1$ (Fig. 6), results in negligibly small $k_{S_1 \to T_1}$ (Supplementary Table 4).

Oppositely to $k_f$, placing the nitro groups at the *ortho* positions yields the largest $k_{ISC}$. For $\textbf{1o}$, $k_{ISC}(S_1^{(CT)} \to T_{(2 \text{ to } j)}) = 5.8 \times 10^{10} \text{ s}^{-1}$ and $k_{ISC}(S_1^{(TICT)} \to T_{(1 \text{ to } j)}^{(t)}) = 9.7 \times 10^{10} \text{ s}^{-1}$. Similarly, for $\textbf{2o}$, $k_{ISC}(S_1^{(CT)} \to T_{(2 \text{ to } j)}) = 5.1 \times 10^{8} \text{ s}^{-1}$ and $k_{ISC}(S_1^{(TICT)} \to T_{(1 \text{ to } j)}^{(t)}) = 3.0 \times 10^{10} \text{ s}^{-1}$. In comparison, the $k_{ISC}$ from the $S_1^{(CT)}$ and $S_1^{(TICT)}$ states of $\textbf{1p}$ is about $3 \times 10^{8} \text{ s}^{-1}$, and of $\textbf{1m}$ – about $8 \times 10^{8}$ $\text{s}^{-1}$ and $2 \times 10^{8} \text{ s}^{-1}$.

Adding extra phenylene linkers drastically increases the ISC rates for $S_1^{(TICT)}$ of the *para* and *meta bis*-nitro-TAPPs, while slightly decreases $k_{ISC}$ for their $S_1^{(CT)}$ states (Table 1). For $\textbf{2m}$ and $\textbf{2p}$, while $k_{ISC}(S_1^{(CT)} \to T_{(2 \text{ to } j)})$ is $1.4 \times 10^{8} \text{ s}^{-1}$ and $4.9 \times 10^{7} \text{ s}^{-1}$, respectively, $k_{ISC}(S_1^{(TICT)} \to T_{(1 \text{ to } j)}^{(t)})$ assumes values that are two orders of magnitude larger, i.e., $1.2 \times 10^{10} \text{ s}^{-1}$ for $\textbf{2m}$ and $4.9 \times 10^{9}$ for $\textbf{2p}$ (Table 1). The latter large rate constants originate from the significant density of triplet states with energy levels closely below $S_1^{(TICT)}$ (Fig. 6). Their contributions to the photophysics of $\textbf{2m}$ and $\textbf{2p}$, however, are negligible to none because the $S_1^{(TICT)}$ states of these compounds lie above the $S_1^{(CT)}$ ones (Fig. 6).

For the *ortho* derivatives, the extra phenylenes have a somewhat opposite effect: a drastic drop in the ISC rates for their $S_1^{(CT)}$ states and only a slight decrease in $k_{ISC}$ for the $S_1^{(TICT)}$ conformers (Table 1). For $\textbf{2o}$, $k_{ISC}(S_1^{(TICT)} \to T_{(1 \text{ to } j)}^{(t)})$ is $3.0 \times 10^{10} \text{ s}^{-1}$, which is only about three times smaller than that for $\textbf{1o}$. Conversely, $k_{ISC}(S_1^{(CT)} \to T_2)$ of $\textbf{2o}$ is about 100 times smaller than that of $\textbf{1o}$ (Table 1). This drastic difference between the ISC rates for the CT states originate from the 20-fold decrease in the SOC between the $S_1^{(CT)}$ and the $T_2$ states induced by the extra phenylenes. Furthermore, the $S_1^{(CT)}$–$T_2$ energy gap is 60 meV for $\textbf{1o}$ and 430 meV for $\textbf{2o}$, and a decrease in the driving force enhances the ISC rates between states with similar geometries, i.e., an "inverted-region" type of behavior[93,94]. Therefore, even though the extra phenylenes improve by five folds the SOC between the $S_1^{(CT)}$ and the low-lying $T_1$ states, while depleting the $S_1^{(CT)}$–$T_2$ SOC, it is the latter, states with similar energies, that has governing effects on $k_{ISC}$.

For the *ortho bis*-nitro-TAPPs $\mathscr{E}(S_1^{(CT)}) > \mathscr{E}(S_1^{(TICT)})$ and the drastic decrease in the $S_1^{(CT)} \to T_2$ ISC rates should not truly benefit the fluorescence properties of $\textbf{2o}$. Nevertheless, $\textbf{2o}$ is weakly fluorescent while $\textbf{1o}$ does not provide any detectable emission (Table 1). For both *ortho bis*-nitro-TAPPs, $k_{ISC}(S_1^{(TICT)} \to T_{(1 \text{ to } j)}^{(t)})$ is more than six orders of magnitude larger than $k_f$ of their $S_1^{(TICT)}$ states. Even possible potential overestimates in the computed $k_{ISC}$ cannot account for such enormous differences. On the other hand, $k_{ISC}(S_1^{(CT)} \to T_2)$ of $\textbf{2o}$ is similar to $k_f$ of its $S_1^{(CT)}$ state, while $k_{ISC}(S_1^{(CT)} \to T_2)$ of $\textbf{1o}$ is about 600 times larger than $k_f$ of its $S_1^{(CT)}$ state. Therefore, while $S_1^{(TICT)}$ states are thermodynamically strongly favored for $\textbf{1o}$ and $\textbf{2o}$, the slowed-down ring rotation, due to the steric hindrance of the nitro groups at the *ortho* position, can provide kinetic traps for some of

the $S_1^{(CT)}$ conformers, resulting in the nanosecond lifetimes extracted from the emission decays of $\textbf{2o}$ (Table 1).

What is the contribution of ISC to the non-radiative deactivation of the *bis*-nitro-TAPPs? For cyclohexane, $\textbf{1p}$ and $\textbf{2p}$ show $\phi_f$ that exceed 0.95 and radiative-decay rate constants that are quite comparable to the TD-DFT computed ones (Table 1). Even assuming the 10% error that tends to be inherent for emission quantum yield measurements, i.e., $\phi_f$ can be as low as 0.9, these experimental findings yield non-radiative decay rate constants, $k_{nd}$, that are smaller than the calculated $k_{ISC}$ for $\textbf{1p}$, which practically is the same for $\textbf{2p}$. These comparisons between experimental and theoretical results indicates some overestimates in the computed ISC rates. Nevertheless, it is safe to assume that ISC dominates the non-radiative deactivation pathways of $\textbf{1p}$ and $\textbf{2p}$ for alkane media, which is consistent with the observed triplet formation in the TA spectra of $\textbf{1p}$ (Fig. 4a). Even a slight increase in the solvent polarity and polarizabilty increases the experimentally measured $k_{nd}$ values, indicating an introduction of IC modes of deactivation. This $k_{nd}$ increase is considerably more pronounced for $\textbf{2p}$ than for $\textbf{1p}$, which correlates with the extra conformational modes aiding the IC. Hence, as the medium polarity increases, the ISC contributions to the deactivation of the excited states diminish.

For $\textbf{1m}$ in cyclohexane, the experimentally measured $k_{nd}$ is quite similar to the computed $k_{ISC}$ for the thermodynamically favored $S_1^{(TICT)}$ state in the gas phase. Nevertheless, ISC does not contribute significantly to the deactivation of $^1\textbf{1m}^*$. The observed emission originates from small amounts of fluorescent transient with 3.7-ns lifetime, with most likely $S_1^{(CT)}$ structure (Table 2). Conversely, the short-lived transients that dominate the TA spectra of $\textbf{1m}$ decay with rates considerably larger than the computed rates of ISC, confirming the assignment of a single character to them (Tables 1 and 2). Considering that $\phi_f = k_f/(k_f + k_{IC} + k_{ISC})$, suggests that $\phi_f^{(CT)}$ of the fluorescent $S_1^{(CT)}$ state of $\textbf{1m}$ cannot exceed 0.34, based on the computed $k_f$ and $k_{ISC}$. In a case of parallel pathways, i.e., $S_1^{(LE)} \longrightarrow S_1^{(CT)} \longrightarrow S_0$ and $S_1^{(LE)} \longrightarrow S_1^{(TICT)} \longrightarrow S_0$ (Table 2), the measured $\phi_f$ is a multiple of $\phi_f^{(CT)}$ and the quantum yield of $S_1^{(CT)}$ formation, which is about 0.2. That is, the measured $\phi_f$ of $\textbf{1m}$ in cyclohexane cannot be larger than 0.068, which is exactly what we observe (Table 1). The maximum value of 0.068 originates when $k_{IC} = 0$ and $k_{nd} = k_{ISC}$, suggesting that both, IC and ISC, are responsible for the non-radiative deactivation of the fluorescent $S_1^{(CT)}$ state of $\textbf{1m}$ in alkane solutions. Still, the dark state, $S_1^{(TICT)}$, mediates the principal deactivation pathway, indicating that IC dominates the decay of $^1\textbf{1m}^*$.

Extending this train of thought to $\textbf{1o}$ based on the computed values for $k_f$ and $k_{ISC}$ (Table 1), suggests that $\phi_f$ of its $S_1^{(CT)}$ state cannot exceed $1.6 \times 10^{-3}$, and for its $S_1^{(TICT)}$ state, $\phi_f \lesssim 2.3 \times 10^{-7}$. That is, even the fluorescent, $S_1^{(CT)}$ state of $\textbf{1o}$ is not all that fluorescent. Nevertheless, $\phi_f = 1.6 \times 10^{-3}$ warrants detectable fluorescence. Assuming, however, that the less favorable $S_1^{(CT)}$ conformer forms with yields that do not exceed 0.1, and considering contribution from IC, can readily place its emission under the detection limits for steady-state spectroscopy.

In contrast, $\textbf{2o}$ presents a slightly different paradigm. With $\phi_f \lesssim 1.4 \times 10^{-6}$, the emission from its $S_1^{(TICT)}$ state is still practically undetectable. The computed $k_f$ and $k_{ISC}$ for its fluorescent state, $S_1^{(CT)}$, on the other hand, suggests that its $\phi_f^{(CT)}$ can be as large as 0.53. The measured values for $\phi_f$ of $\textbf{2o}$ are about 1,000 times smaller (Table 1), which is consistent with minute quantum yields of $S_1^{(CT)}$ formation along with contributions from IC modes of deactivation. Nevertheless, these findings for $\textbf{2o}$ show an example where a 100-fold decrease in $k_{ISC}$, induced by extra molecular flexibility and decoupling with the nitro groups, can produce detectable fluorescence.

The presented results so far focus solely on calculations that, while describing well the experimental trends for non-polar solvents, do not provide insight into the polarity-induced changes in the photophysics of the *bis*-nitro-TAPPs. The CT character of the photoemissive states, $S_1^{(CT)}$, unequivocally explains the positive solvatofluorochromism of the fluorescent *bis*-nitro-TAPPs. The polarity-induced bathochromic shifts in the emission spectra, however, accompany a drastic decrease in $\phi_f$ to an extent where neither of the *bis*-nitro-TAPPs, except **2o**, fluoresces in polar media. The TICT states, with disrupted π-conjugation between the donors and the acceptors, have a stronger CS character than the CT states with substantial spatial overlap of the transition orbitals (Supplementary Fig. 10). Therefore, an increase in solvent polarity has stronger effects on the $S_1^{(TICT)}$ than on the $S_1^{(CT)}$ states, causing a decrease in the energy gaps for the IC transitions, $S_1^{(TICT)} \rightarrow S_0^{(TICT,FC)}$. For fluorophores with $S_1$-$S_0$ gaps smaller than 2.5 eV, i.e., for green-red emitters, IC becomes a substantial contributor to the non-radiative deactivation channels[95–98]. In addition, even for blue fluorophores IC can strongly quench the fluorescence[96,98]. Thus, it is not surprising that a strong decrease in $\phi_f$ of **1p**, **1m**, **2p**, and **2m** accompanies the positive solvatochromic effects in their emission.

To examine the extent of these effects on **1p**, we calculate IC rates for the $S_1^{(TICT)} \rightarrow S_0^{(TICT,FC)}$ transition. An increase in media polarity pushes the fluorescence maxima of **1p** to about 600 nm (Fig. 6). Therefore, we decrease the $S_1$-$S_0$ gap from 1.98 to 1.68 eV, with increments of 0.06 eV, to test its effect on $k_{IC}$; and employ FC and HT formalisms for computing $k_{IC}$. In contrast to the fluorescence from $S_1^{(CT)}$, the FC transition from $S_1^{(TICT)}$ to $S_0^{(TICT,FC)}$ is symmetry forbidden. Conversely, a decrease in the $S_1$-$S_0$ gap causes an exponential increase in the IC-HT rates to produce a $k_{IC-HT}$ value in the order of $10^9$ s$^{-1}$ at 1.8 eV. This finding suggests an efficient quenching of the **1p** fluorescence as its emission bathochromically shifts to wavelengths in the range of 600–700 nm, which is in an excellent agreement with the experimental observations. At the same time, $k_{IC-FC}$ values are three orders of magnitude smaller than $k_{IC-HT}$, and similarly to a free-base porphyrins[98], the IC-FC pathways do not significantly contribute to the total IC rates. This theoretical reproduction of the experimental trends with the HT (rather than FC) formalism has a key implication about the importance of vibronic borrowing for efficient IC in these *bis*-nitro-TAPPs.

The theoretical analysis shows that slowing down ISC and making $k_{ISC}$ comparable with or smaller than $k_f$ are key for attaining fluorescence from organic dyes with nitro substituents. The CT character, inherent for nitro-aromatics, plays an important role in the solvent-polarity effects on their photophysics. The extent of CS is crucial for balancing between the Born and Onsager solvation stabilization of the excited states and lowering some of them too close to the $S_0$ PES. These unequal effects of medium polarity on the $S_1$-$S_0$ gaps at the different minima of the $S_1$ PES have drastic implications on the observed photophysics. As the $S_1$-$S_0$ gaps decrease, vibronic borrowing starts to play a crucial role in the non-radiative IC deactivation.

## Discussion

To make nitro derivatives of organic chromophores fluorescent, it is essential to: (1) enhance the rates of $S_1 \longrightarrow S_0$ radiative decay; (2) suppress ISC; and (3) slow down the IC from $S_1$ to $S_0$. The well-known propensity of nitro groups to induce triplet formation has placed the focus of this line of research on making ISC rates slower than, or at least comparable to, the rates of radiative decay. For most *bis*-nitro-TAPPs, however, $k_{ISC}$ is comparable to $k_f$, and IC becomes deterministic for their photophysics.

Weakening the coupling with the nitro groups by placing them on phenyl substituent offers a means for dropping $k_{ISC}$ below $10^9$ s$^{-1}$, which is crucial because the fluorescent states of most organic chromophores have nanosecond lifetimes. Extra decoupling with the nitro groups, by adding phenylene linkers, further decreases $k_{ISC}$ for the excited states with symmetric geometries, i.e., $S_1^{(CT)}$. For asymmetric structures, such as the $S_1^{(TICT)}$, however, additional degrees of conformational freedom increase the density of triplet states with energy levels slightly below $S_1$. Thus, considering the evolution of the molecular symmetry and the balance between these two opposing effects from extra phenylene spacers is essential when undertaking approaches for slowing down the ISC rates.

An important trend that emerges from the *bis*-nitro-TAPPs is the effects of the dihedral angles, $\alpha$, with the nitrophenyls, which correlate well with the observed $\phi_f$ (Supplementary Table 3). Moving the nitro groups from *para* to *meta* weakens their electronic coupling with the pyrrolopyrrole core and results in a slight increase in $\alpha$ of $S_0$, e.g., from 32º for **1p** to 38º for **1m**. In the $S_1^{(CT)}$ states, $\alpha$ relaxes to about 20º for both, **1p** and **1m** (Supplementary Fig. 10). Conversely, placing the nitro group in the ortho position, induces steric hindrance and locks conformers with large $\alpha$, i.e., 50º for $S_0$ and 43º for $S_1^{(CT)}$ of **1o** (Supplementary Fig. 10). These structural features correlate quite distinctly with the measured non-radiative decay rates and with the calculated rates of ISC (Table 1), i.e., orthogonality between the ring system of the donor and of the nitro-bearing acceptor improves the SOC between closely lying $S_1$ and $T_j$ states (Supplementary Fig. 10). While for **1p** and **1m** $k_{ISC} < 10^9$ s$^{-1}$, for **1o** $k_{ISC} > 5 \times 10^9$ s$^{-1}$ (Table 1). Introducing orthogonality between the aromatic rings appears to enhance SOC and the ISC rates.

An increase in the dihedral angles, $\alpha$, decreases the electronic coupling with the nitro groups needed for slowing down ISC. Orthogonality between the rings, on the other hand, provides a means for improved overlaps between orbitals with different momenta, such as π and σ, which can have dominating effects on enhancing SOC[99]. These finding suggest that placing the nitro groups on aromatic rings that are linked with the principal chromophore *via* σ-bonds is essential for decreasing $k_{ISC}$. To maintain favorably low ISC rates, however, these structures should be kept rigid and with some degree of planarity.

As expected, keeping rigidity of the structures in order to supress ring rotation leads to an IC decrease of the non-radiative decay rates (Table 1). Rigidifying structures with orthogonal dihedral angles, on the other hand, can dramatically enhance ISC.

Enhancing the radiative-decay rates represents another approach for making nitro compounds fluorescent. Examining the photophysics of the *bis*-nitro-TAPPs shows values of $k_f$ that do not exceed $10^9$ s$^{-1}$ (Table 1), which is common for organic chromophores, set by inherent limits. Conversely, for the non-fluorescent *bis*-nitro-TAPPs, the calculated values of $k_f$ drop to about $10^4$ s$^{-1}$. This trend warrants rewording of the above stated aim. Instead of asking how to enhance $k_f$, it is beneficial to focus on how to prevent $k_f$, inherent for a chromophore, from decreasing upon adding nitro groups. The strong electron-withdrawing propensity of nitro substituents leads to excited states with a pronounced CT character. An increase in the extent of CS reduces the overlap between the frontier orbitals and the transition dipole moment for radiative deactivation. The orthogonality between the pyrrolopyrrole core and the nitrophenyls improves the separation between the singly-occupied orbitals of the CT excited states. That is, orthogonality decreases the orbital overlap and diminished the transition dipole moments. Indeed, the smallest $k_f$ values are attributed to the $S_1^{(TICT)}$ structures with highest orthogonality (Table 1). Therefore, maintaining some planarity to allow overlapping between the singly occupied orbitals of the CT excited states ensures reasonably large radiative-decay rates essential for making nitro compounds fluoresce.

Excited states with pronounced CT character are inherent for nitro derivatives of organic compounds. Considering orbital-density distributions indicates that differences in the CT characters of the $S_1$ and $T_j$ states suppresses the ISC between them[19]. At the same time polar solvents stabilize states with a CS character and lower their energy levels. When the ground state is not as polar as the $S_1$ state with a CT character, an increase in medium polarity narrows the $S_1$-$S_0$ energy gap and induces pathways for efficient IC. Such IC to a non-polar ground state can be view as charge recombination (CR). The molecular structure of most *bis*-nitro-TAPPs ensure slow ISC while maintaining reasonably large $k_f$, which is reflected by their strong fluorescence in non-polar media. An increase in solvent polarity, however, allows for CR-mediated IC to take precedent and quench the observed emission. As revealed by its fluorescence and a lack of solvatochromism, **2o** shows a means for supressing this polarity-induced fluorescence quenching. The small difference between the permanent dipoles of the ground and excited states of **2o**, ensures a similar septicity to solvent polarity. That is, an increase in medium polarity does not stabilize $S_1$ more than $S_0$, which keeps their PESs sufficiently apart to supress dominating IC.

Placing the nitro groups at *ortho* positions makes **1o** completely non-fluorescent. Indeed, **1o** also manifests the largest $k_{ISC}$, but as TA analysis shows, ISC cannot account for the picosecond and sub-picosecond deactivation of **1o** in non-polar media. That is, CR-mediated IC is responsible for the quenching of the emission of **1o**. In addition, the CT transitions mediated by **1o** most likely involve multiple pathways. The efficiency of through-bond CT manifests smaller falloff with distance than through-space interactions. Therefore, the discussion focuses on through-bond coupling with the nitro groups. When the nitro groups are at the *ortho* position, however, they are spatially close to the pyrrolopyrrole donor. This proximity makes the efficiency of CT through such direct van der Waals contacts as efficient as the through-bond processes. Such extra electronic-coupling pathways explain the sub-picosecond and picosecond kinetics of the excited-state CT and CR IC to the ground state mediated by **1o**, which appears fast for similar systems with a single σ-bond linking the donor with the acceptor[100].

While decreasing the ISC rates for the *bis*-nitro-TAPPs makes them fluorescent in non-polar solvents, it also allows the effects from IC to emerge and dominate their photophysics for polar media. CT transitions are inherent for the excited-state dynamics of organic compounds bearing nitro groups. Efficient CR, driving the IC to the ground state, is a principal source of fluorescence quenching induced by the medium polarity.

The kinetics of CR and CT, in general, has a complex dependence on solvent polarity, which affects the electrochemical potentials of the donor and the acceptor, the Coulombic work term, and the medium reorganization energy, $\lambda_o$[94,101]. Nevertheless, increasing the sizes of the donor and the acceptor can suppress the polarity-induced lowering of $S_1$ states with a CS character. Expanding the delocalization of the positive charge on the oxidized donor and the negative charge on the nitro-bearing reduced acceptor not only improves the orbital overlaps needed for large transition dipole moments for radiative deactivation, but also decreases the susceptibility of the CT state to the solvent polarity. As implemented by the Born solvation energy, polarity-induced stabilization of CT states is inversely proportional to the sizes of the comprising charged species[94,102,103].

The *bis*-nitro-TAPPs studied herein represent an exceptional case as a result of linking a particularly strong electron donor with good electron acceptors *via* a biaryl motifs. Our study demonstrates that this combination leads to large fluorescence quantum yields as long as the orbitals carrying the positive charge of the oxidized donor and the negative charge of the reduced acceptor in

the CT states overlap well. Such combination translates to large $k_f$, the CT character of the excited states and substantially reduced the rates of ISC. Concurrently, this CT character makes the fluorescence immensely sensitive to medium polarity.

Placing nitro groups on phenyl substituents ensures electronic coupling with the principal chromophore core that is (1) sufficiently weak to ensure slow intersystem crossing and intense fluorescence, and (2) sufficiently strong to introduce substantial charge-transfer character that controls the excited-state dynamics. Along with the medium polarity, moving the nitro groups to different positions of the phenyl substituents changes the rates of internal conversion with many orders of magnitude. The internal conversion, occurring via back charge transfer, strongly relies on vibronic borrowing which increases its rates with orders of magnitude. When the fluorescence is strong, intersystem crossing dominates the non-radiative deactivation channels. Conversely, internal conversion is the primary pathway for deactivation of the weakly fluorescent or non-fluorescent cases. This mechanistic balance between radiative, internal-conversion and intersystem crossing channels reveals structure-function relationships that are crucial for the design of fluorescent nitro-aromatics and[104] the future developments of *n*-type organic conjugates for photonics, electronics and energy science and engineering.

## Methods

**Synthesis**. For the synthesis and characterization of the *bis*-nitrotetraphenylpyrrolopyrroles, **1p**, **1m**, **1o**, **2p**, **2m**, and **2o** see Supplementary Methods (Synthetic procedures and $^1$H and $^{13}$C NMR spectra for synthesized compounds)

**Photophysics**. For the steady-state absorption and emission experimental details see Supplementary Methods (Steady-state absorption and emission). For the time-resolved fluorescence details see Supplementary Methods (Time correlated single photon counting experiments). For the time-resolved pump-probe transient absorption details see Supplementary Methods (Transient-absorption spectroscopy).

**Computations**. For the computational analysis details, employing time-dependent density-functional theory: see Supplementary Methods (Computational details).

## Data availability

The main data supporting the findings of this study are included in the paper, its Supplementary Information and Supplementary Data files 1–6. Additional raw data i.e. optical and NMR spectra a long with atomic coordinates of the dyes are available here: https://doi.org/10.6086/D13X22.

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

## Acknowledgements

The authors would like to thank the Foundation for Polish Science (TEAM POIR.04.04.00-00-3CF4/16-00) and Global Research Laboratory Program (2014K1A1A2064569) through the National Research Foundation (NRF) funded by Ministry of Science, ICT & Future Planning (Korea), and the U.S.A. National Science Foundation (grant CHE 1800602). The theoretical calculations were performed with computational resources provided by the High Performance Computing Center North (HPC2N) which is a Swedish national center for Scientific and Parallel Computing through the project "Multiphysics Modeling of Molecular Materials" SNIC 2019-2-41. G.B. and H.Å. thank the support of Carl Tryggers foundation (Grant No. CTS 17:537), Olle Engkvist Byggmästare foundation (Contract No. 189-0223).

## Author contributions

The manuscript was achieved through contributions from all authors. Y.P. designed the project and conducted the synthetic experiments. G.I.B. and I.D. conducted computational studies. E.M.E. and J.A.C. performed the photophysical studies. All authors wrote the manuscript. D.T.G., V.I.V., and H.Å. supervised the project.

## Competing interests

The authors declare no competing interests.
