## [Peer Review File · Communications Chemistry]

Reviewers' comments:

Reviewer #1 (Remarks to the Author):

GENERAL COMMENTS

In this article, the authors present a series of theoretical and experimental studies for a set of nitro-TAPP derivatives based on acceptor-donor-acceptor structures.

The results are very interesting and allow us to better understand the existence of measurable fluorescence in these compounds, which is a very relevant topic. In my opinion, this manuscript is suitable for publication in Communications Chemistry after minor corrections.

Some questions, comments, and suggestions to the authors are presented below to improve the manuscript.

QUESTIONS, COMMENTS, AND SUGGESTIONS

1. The title is very general. Once only TAPP-based structures were studied, it could be interesting to mention that it is a case study for pyrrolo[3,2-b]pyrrole based materials, otherwise the hypotheses presented in the manuscript should be validated for other nitroaromatic compounds, which is not the case.
2. The discussions presented in the subsections "Rationale for the molecular designs" and "Solvent effects on the optical spectra" should be reinforced or based on the theoretical results (those presented already presented in the Supplementary Material and additional ones), for instance:
 - 2-i) the steric hindrance imposed by the nitro group could be compared to the structural features of the GS (optimized) geometries;
 - 2-ii) the electronic coupling between A-D-A units could be explored from the evaluation of the partial density of states of these blocks and via the spatial distribution of the frontier molecular orbitals;
 - 2-iii) the arguments associated with the "planarity between pyrrolo[3,2-b]pyrrole and the benzene" must be better explained once it is not completely in line with the theoretical data;
 - 2-iv) it is not clear why the theoretical absorption spectra are not presented and compared with the experimental data. The evaluation of the theoretical results (transitions, fosc, etc) could improve the discussion and help to understand the nature of the transitions observed in the systems (in particular for 1m and 1o), as well as the molecular orbitals involved. In addition, it could be identified the existence of additional transitions in 2o that are apparently superimposed in Fig. 2 f;
 - 2-v) the "extension of the pi-conjugation over the additional phenylenes" could be better investigated in comparison with the theoretical results. Indeed an improvement of the conjugation is evident only in the HOMO of the 1p molecule.
3. Please revise the statement "The emission spectra of the fluorescent bis-nitro-TAPPs, manifest strong positive solvatochromism", once it is not the case of 2o.
4. Please, provide references to the following statements: "The nitro group, indeed, provides a high density of isoenergetic triplet and singlet excited states with different symmetries, offering multiple paths for electron transitions between orbitals with different momenta, such as n and π -bonding, that allows spin flip and ISC, i.e., a classical El Sayed mechanism".
5. The data shown in Figure 3 must be explained, referenced, and discussed in more detail.
6. On page 8 the results related to DMSO are associated with Figure 4e, however, it is linked to cyclohexane. On this same page (paragraph 4) it is mentioned results associated with toluene, however, it is not shown.

7. The caption of Figure 4 could be improved, and the most relevant parameters coming from TA analysis should be organized in a table (probably in the Supplementary material)
8. Please, consider revising the text of pages 6 and 8 of the pdf file, they are a little bit repetitive. Example: "Similar to 1p... Similar to 1p" (page 8).
9. Regarding "Computational analysis", I point out the following considerations:
- 9-i) Given the relevance of S1(TICT) excited state in the model, more details regarding how the scan of the S1 PES was conducted and on the evaluation of the minima (vibrational analysis) could be provided. Some general aspects of the theory are presented in the Suppl. Material, however more specific information on how the calculations were conducted should be provided.
- 9-ii) Most of the discussions are based on the oscillator strengths coming from TD-DFT calculations, however, it is known that this approach presents some limitations to describe the effective coupling between excited states in comparison to MRCI. It should be discussed.
- 9.iii) It is known that CT excited states are problematic for TD-DFT when using standard xc-functionals, which allow us to understand the limited accuracy of the B3LYP-based approach (that could be referenced in Suppl Mat). Indeed, the accuracy of TD-DFT to describe such excited states (including dark states ordering) can be achieved by comparing the results coming different functionals (in particular long-range-corrected or hybrid xc-functionals with different amounts of HF exchange). In this sense, I consider that the partial results of the optical absorption spectra of the systems should be provided and compared with the results presented in Figure 2 of the MS (at least those coming from TD-DFT/B3LYP and TD-DFT/CAM-B3LYP). Also, the above-mentioned limitations should be better discussed.
- 9.iv) All the calculated transition rates could be summarized in tables to ease the interpretations.
10. In its current form, it is clear that the manuscript was divided into a series of separated analyses. It could be interesting to discuss the results in a more concatenated way (as exemplified in Comment 2).
11. Finally, based on the obtained results, is it possible to propose a set of (practical) rules for the design of new fluorescent nitro-aromatics? Indeed it was stated in the conclusions "This mechanistic balance between radiative, internal-conversion and intersystem crossing channels reveals structure-function relationships that are crucial for the design of fluorescent nitro-aromatics and the development of new n-type organic conjugates for photonics, electronics and energy science and engineering", however, it is not so clear, and must be better discussed.

Reviewer #2 (Remarks to the Author):

The paper of D.T. Gryko et al deals with the photobehaviour of six quadrupolar acceptor-donor-acceptor dyes with nitrophenyl as acceptor groups and a pyrrolo[3,2-b]pyrrole as central donor unit, differing for the position of the nitro groups with respect to the central core (para, meta and ortho) or by the insertion of a phenylene linker between donor and acceptor moieties. The challenging purpose of the authors, underlined by the title of the paper, is to definitively understand the reason of the unusual fluorescence of nitro-aromatics to discover important structural figures to drive the synthesis of nitro-compounds suitable for molecular photonics and electronics.

However, the manuscript is difficult to read and to understand.

I am sorry to suggest rejection of this ms in its present form and resubmitting a revised version taking into account the following criticism

1. I suggest to rewrite the paper separating the new results from their interpretation that is often repeated along the text in a messy way. The Results section should be better organized and separated from the Discussion section. The data have to be collected in tables and not reported

scattered along the text and mixed to the interpretation. For instance, the table containing the fluorescence data has to be shown in the text and not in the Supporting Information if the data are new; a table reporting all the fs transient absorption results (the transients with their properties as lifetime, peak position and the assignment of the transients in the last column) should be introduced. Also the calculations results are not clearly reported: a table containing in detail the calculated energies for the singlet and triplet states with the nature and the oscillator strength of the transitions (sketched in Figure 6) should be added in the Supporting Information. A comparison between experimental and calculated electronic spectra is missing. In fact, the discussion and interpretation of the absorption spectra is weak in absence of any support from calculated transitions.

The figures should be magnify and clarify, mainly Figure 2 and 4, to allow also the reader to observe the obtained results.

2. The assignment of the TA transients to emitting and dark states seems sometimes arbitrary in absence of fs up-conversion experiments.

3. One important item of the present paper is the competition between fluorescence and intersystem crossing in the six nitro-derivatives. For this purpose, first of all the experimental derived kF should be clearly reported in a table and compared with the calculated values to support the reliability of the calculations. As to the kISC and the eventual presence of significant population of the triplet state the authors completely rely on the calculations even if, in some points along the text, they admit that the calculations appear to overestimate kISC. In my opinion the measure of the triplet formation yield for the investigated nitro-compounds in different solvents is necessary to a reliable comparison.

4. The main purpose of this study would be to suggest structural design for nitro-aromatics leading to high fluorescent systems. However, in the Conclusions section it is not clear what are the strategies for obtaining fluorescent nitro derivatives. It seems that the only suggestion is to placing nitro groups on the phenyl substituents but I know many examples of nitrophenyl derivatives that show different emissive behaviour. Please better summarize your results.

Reviewer #3 (Remarks to the Author):

The manuscript describes six bis-nitroterraphenylpyrrolopyrrole acceptor-donor-acceptor systems

The main result of this contribution is illustrating the understanding of the the large variability of the photophysical behavior of these systems as a function of solvent-environment, substituent positions. These systems are excellent models for the development of fluorophores where the nitro group plays an essential role and implies the presence of several photophysical channels. The model to study these effects correspond to a series of dihydropyrrolo-pyrroles systematically substituted with nitrophenyl groups.

The introduction is excellent and gives a broad understanding of the effects of the NO₂ group in chromophores.

Among the six molecules of this study, only the one with the NO₂ group in the ortho position with a single phenyl bridge is non-fluorescent. The emission of these systems undergoes red-shifting with increasing the solvent polarity and a reduction in the emission yield. Upon increasing bridge of these systems with a n additional phenyl group, the molecules have a relative increase in the Stokes shift. When the systems are placed in a rigid solid environment, there is an important increase in the emission yields and a reduction in the Stokes shifts.

The study also includes TDDFT calculations that indicate that there are two types of pathways for the locally excited states. One that forms a CT state and another one that forms a TICT-type

(dark) state. Another important finding of this contribution is that from the electronic structure of these molecules, the SOC matrix elements and intersystem crossing rates are particularly small in comparison with nitrated polycyclic aromatic hydrocarbons where the isc rates can be as large as 10^{13} s^{-1} .

I consider that this manuscript signifies an important contribution with clear results about an important topic. I recommend publication after the following minor issues are considered by the authors:

-In the introduction the authors should consider mentioning the possibility of the effect of the nitro group dihedral angle with respect to the aromatic systems, where, for some nitroaromatics with highly twisted NO_2 groups, tend to undergo nitric oxygen dissociation.

-In the section about solid solvent medium, the possibility for formation of CT states in these media is not mentioned explicitly. I think the readers will be expecting that this topic is included or at least commented more explicitly in this section.

-I do suggest that in addition to Figure 6, the authors attempt to summarize their results in a graphical scheme or simple table which indicates increases and decreases in the non-radiative and radiative rate constants depending on the NO_2 position (and the reason, in terms of the respective state shifting), presence of an additional phenyl bridge, or the solvent environment, and use of a solid environment; otherwise it is a bit difficult to quickly find a specific effect for a given molecule.

-The same kind of comment would apply to the conclusions section. The manuscript is quite thorough in its discussions in the results section. A summary of the effects for each molecule should be found in the conclusions section.

Point-by-point response

Reviewer 1.

In this article, the authors present a series of theoretical and experimental studies for a set of nitro-TAPP derivatives based on acceptor-donor-acceptor structures. The results are very interesting and allow us to better understand the existence of measurable fluorescence in these compounds, which is a very relevant topic. In my opinion, this manuscript is suitable for publication in Communications Chemistry after minor corrections. Some questions, comments, and suggestions to the authors are presented below to improve the manuscript.

We would like to thank the reviewer for evaluating our work positively and for recognizing the importance of our work. Below we provide answers to the important points raised by the reviewer.

Comment 1. The title is very general. Once only TAPP-based structures were studied, it could be interesting to mention that it is a case study for pyrrolo[3,2-b]pyrrole based materials, otherwise the hypotheses presented in the manuscript should be validated for other nitroaromatic compounds, which is not the case.

Response: We replaced the original title i.e. 'Deciphering the Enigma of Unusual Fluorescence in Weakly Coupled Nitroaromatics' with 'Deciphering the Enigma of Unusual Fluorescence in Weakly Coupled Bis-nitro-pyrrolo[3,2-*b*]pyrroles

Comment 2. The discussions presented in the subsections "Rationale for the molecular designs" and "Solvent effects on the optical spectra" should be reinforced or based on the theoretical results (those presented already presented in the Supplementary Material and additional ones), for instance:-i) the steric hindrance imposed by the nitro group could be compared to the structural features of the GS (optimized) geometries;

Response: We have introduced appropriate discussion of structural features for ortho-substituted 1a and 2o molecules in comparison with *para* and *meta* representatives.

Comment 3.-ii) the electronic coupling between A-D-A units could be explored from the evaluation of the partial density of states of these blocks and via the spatial distribution of the frontier molecular orbitals;

Response: we are thankful to referee for this comment. That is indeed possible to do, but requires quite a lot of supplementary calculations and resulting data will not provide some new vision of photophysical behavior of the studied species. In our opinion the key qualitative conclusions about electronic coupling between A-D-A units can be easily obtained from Fig. S11. Comparing the Fig. S11 with oscillator strength values in Figure 6 and with the calculated k_r values in Figures S9 and S10 one can easily establish a correlation between the efficiency of electronic coupling between A-D-A units and emission efficiency.

Comment 4. the arguments associated with the "planarity between pyrrolo[3,2-b]pyrrole and the benzene" must be better explained once it is not completely in line with the theoretical data;

Response: We have introduced an additional discussion associated with the mutual interposition of pyrrolo[3,2-b]pyrrole and the benzene vs. coupling efficiency between D-A fragments.

Comment 5. it is not clear why the theoretical absorption spectra are not presented and compared with the experimental data. The evaluation of the theoretical results (transitions, fosc, etc) could improve the discussion and help to understand the nature of the transitions observed in the systems (in particular for 1m and 1o), as well as the molecular orbitals involved. In addition, it could be identified the existence of additional transitions in 2o that are apparently superimposed in Fig. 2 f;

Response: we have additionally presented the absorption spectra of all studied species in Supplementary Figure 9. The detailed info on electronic transitions were also added into Supplementary Materials (Tables 2-7). Our quantum-chemical calculations generally well reproduce the absorption spectra of all studied molecules demonstrating a slight systematic overestimation of electron transition energies. The satellite absorption bands in high-energy region for some molecules also well reproduced in our calculations and the orbital nature of these transitions as well as assignment of all absorption bands in calculated spectra can be found in Supplementary Figures 9 and 10 and in Supplementary Tables 2-7.

Comment 6. the “extension of the pi-conjugation over the additional phenylenes” could be better investigated in comparison with the theoretical results. Indeed an improvement of the conjugation is evident only in the HOMO of the 1p molecule.

Response: we have introduced an additional discussion of the “extension of the pi-conjugation over the additional phenylenes” in 2o, 2m and 2p species in terms of rotation degree between pyrrolo[3,2-b]pyrrole core and adjacent benzene rings (last paragraph in section “Optical absorption spectra”).

Comment 7. Please revise the statement “The emission spectra of the fluorescent bis-nitro-TAPPs, manifest strong positive solvatochromism”, once it is not the case of 2o.

Response: We thank the reviewer for catching it up. We revised this statement to “The emission spectra of the fluorescent *bis*-nitro-TAPPs, except of dye **2o**, manifest strong positive solvatochromism”

Comment 8. Please, provide references to the following statements: “The nitro group, indeed, provides a high density of isoenergetic triplet and singlet excited states with different symmetries, offering multiple paths for electron transitions between orbitals with different momenta, such as n and π -bonding, that allows spin flip and ISC, i.e., a classical El Sayed mechanism”.

Response: We added a couple of references that support this statement.

Comment 9. The data shown in Figure 3 must be explained, referenced, and discussed in more detail.

Response: The data presented on Figure 3 has been explained and discussed on over two additional pages of text.

Comment 10. On page 8 the results related to DMSO are associated with Figure 4e, however, it is linked to cyclohexane. On this same page (paragraph 4) it is mentioned results associated with toluene, however, it is not shown.

Response: The reviewer is correct. Figure 4e in the original manuscript shows TA results for DCM, not DMSO. Nevertheless, the original text states “--- polar solvents --- for DCM and DMSO (Figure 4e).” i.e., Figure 4e is an example illustrating the trends for one of the solvents. The results for the other

one are quite similar and for the sake of brevity we did not present it. Therefore, we revised Figure 4, placed additional TA results in the ESI and edited the text accordingly.

Comment 11. The caption of Figure 4 could be improved, and the most relevant parameters coming from TA analysis should be organized in a table (probably in the Supplementary material)

Response: The reviewer is correct to request this information. We revised the caption of Figure 4 and added extra experimental details to the ESI.

Comment 12.. Please, consider revising the text of pages 6 and 8 of the pdf file, they are a little bit repetitive. Example: "Similar to 1p... Similar to 1p" (page 8).

Response: We agree with the reviewer and revised the text to eliminate the repetitions of the same wording in sentences that are near one another.

Comment 13.. Regarding "Computational analysis", I point out the following considerations:-i) Given the relevance of S₁(TICT) excited state in the model, more details regarding how the scan of the S₁ PES was conducted and on the evaluation of the minima (vibrational analysis) could be provided. Some general aspects of the theory are presented in the Suppl. Material, however more specific information on how the calculations were conducted should be provided.

Response: at initial stage we optimized the S₁ and T₁ states of all studied dyes by using TDDFT/B3LYP and UB3LYP methods. Usually, S₁ and T₁ are characterized by a similar geometry because of they are usually of the same electronic configuration (HOMO-LUMO type in most cases). In vertical absorption spectra (starting from S₀ ground state geometry) we really see that S₁ and T₁ are of the same configuration, but optimized S₁ and T₁ states are very different in structure. T₁ state is closer to the ground state conformation, while S₁ global minimum (TICT) demonstrate a strong rotation along D-A pairs and not demonstrate a fluorescence. Thus, we decided that it should be additional "bright" local minimum on S₁ PES. Starting from T₁ state geometry and optimizing it by TDDFT/B3LYP we obtained S₁^(CT) bright state which is very close to T₁ state geometry and demonstrate a strong fluorescence (especially for 1p and 2p). In all cases (T₁, S₁^(CT), S₁^(TICT)) we checked the vibrational spectra of the optimized structures. No imaginary frequencies were found meaning that these geometries correspond to the genuine PES minima. Corresponding discussion was added to "Computational details" section in supplementary materials file.

Comment 14. Most of the discussions are based on the oscillator strengths coming from TD-DFT calculations, however, it is known that this approach presents some limitations to describe the effective coupling between excited states in comparison to MRCI. It should be discussed.

Response: MRCI approach is known to be very computationally expensive in sense of required time and resources. Even for simple molecules containing up to 10 atoms MRCI calculations are quite challenging. At the current state of computational chemistry progress the MRCI calculations in its classical implementation are impossible to perform for such large systems like TAPPs. The applicability of TDDFT in case of studied dyes can be explained by a good agreement of the calculated data with experimental spectra (both absorption and fluorescence).

Comment 15. It is known that CT excited states are problematic for TD-DFT when using standard xc-functionals, which allow us to understand the limited accuracy of the B3LYP-based approach (that

could be referenced in Suppl Mat). Indeed, the accuracy of TD-DFT to describe such excited states (including dark states ordering) can be achieved by comparing the results coming different functionals (in particular long-range-corrected or hybrid xc-functionals with different amounts of HF exchange). In this sense, I consider that the partial results of the optical absorption spectra of the systems should be provided and compared with the results presented in Figure 2 of the MS (at least those coming from TD-DFT/B3LYP and TD-DFT/CAM-B3LYP). Also, the above-mentioned limitations should be better discussed.

Response: we have added the absorption spectra calculated by TD-DFT/CAM-B3LYP approach for all studied molecules into Supplementary Materials file (Figure S9). TD-DFT/B3LYP calculations strongly underestimate the energies of CT states (see in table below) and we didn't present these results in MS and in supplementary data. The limitation of B3LYP approach is discussed in "Computational details" part in SI file. As requested by Referee, we also tested a performance of another long-range-corrected wb97XD functional for key TAPPs representatives 1o and 1p. Both CAM-B3LYP and wb97XD provides very similar transition energies and oscillator strength (table below), thus the usage of CAM-B3LYP method in our study is reasonable in comparison with other functionals. Slight overestimation of excited state energies by CAM-B3LYP functional is also in a reasonable range.

	CAM-B3LYP	wB97XD	B3LYP
1o	396 (0.3) [exp.420]	384 (0.3)	564 (0.13)
1p	401 (1.9) [exp. 469]	386 (1.7)	518 (1.2)

Comment 16. All the calculated transition rates could be summarized in tables to ease the interpretations.

Response: experimental photophysical characteristics of *bis*-nitro-TAPPs obtained from optical emission and steady-state absorption studies are now summarized in a separate table. Theoretically calculated rate constants are summarized in Figures 11 and 12 (in Supplementary).

Comment 17. In its current form, it is clear that the manuscript was divided into a series of separated analyses. It could be interesting to discuss the results in a more concatenated way (as exemplified in Comment 2).

Response: The reviewer brings a valid point. Therefore, we added a "Discussion" section that brings together the results from the different types of studies and outlines their implications for the fluorescence of the nitro derivatives.

Comment 18. Finally, based on the obtained results, is it possible to propose a set of (practical) rules for the design of new fluorescent nitro-aromatics? Indeed it was stated in the conclusions "This mechanistic balance between radiative, internal-conversion and intersystem crossing channels reveals structure-function relationships that are crucial for the design of fluorescent nitro-aromatics and the development of new n-type organic conjugates for photonics, electronics and energy science and engineering", however, it is not so clear, and must be better discussed.

Response: In the new "Discussion" section, we outline the general trends and recommendation, which originate from this study, for making nitro-compounds fluoresce.

Reviewer 2.

The paper of D.T. Gryko et al deals with the photobehaviour of six quadrupolar acceptor-donor-acceptor dyes with nitrophenyl as acceptor groups and a pyrrolo[3,2-b]pyrrole as central donor unit,

differing for the position of the nitro groups with respect to the central core (para, meta and ortho) or by the insertion of a phenylene linker between donor and acceptor moieties. The challenging purpose of the authors, underlined by the title of the paper, is to definitively understand the reason of the unusual fluorescence of nitro-aromatics to discover important structural figures to drive the synthesis of nitro-compounds suitable for molecular photonics and electronics. However, the manuscript is difficult to read and to understand. I am sorry to suggest rejection of this ms in its present form and resubmitting a revised version taking into account the following criticism *We would like to thank the reviewer for evaluating our work positively. Below we provide answers to the important points raised by the reviewer.*

Comment 1. I suggest to rewrite the paper separating the new results from their interpretation that is often repeated along the text in a messy way. The Results section should be better organized and separated from the Discussion section. The data have to be collected in tables and not reported scattered along the text and mixed to the interpretation. For instance, the table containing the fluorescence data has to be shown in the text and not in the Supporting Information if the data are new; a table reporting all the fs transient absorption results (the transients with their properties as lifetime, peak position and the assignment of the transients in the last column) should be introduced. Also the calculations results are not clearly reported: a table containing in detail the calculated energies for the singlet and triplet states with the nature and the oscillator strength of the transitions (sketched in Figure 6) should be added in the Supporting Information. A comparison between experimental and calculated electronic spectra is missing. In fact, the discussion and interpretation of the absorption spectra is weak in absence of any support from calculated transitions. The figures should be magnify and clarify, mainly Figure 2 and 4, to allow also the reader to observe the obtained results.

Response: The reviewer brings a range of valid points.

We reorganized the presentation of the “Results”, and added a new “Discussion” section.

A new Table 1 summarizes the optical absorption and emission results, along with the radiative and non-radiative rate constants extracted from the experimental measurements.

Presenting in Table 1 some of the rate constants obtained from the TD-DFT calculations allows for visualization of the comparisons between experiment and theory.

A new Table 2 summarizes the spectral features of the different transients and their kinetics.

We have also improved the flow of the description of computational results.

Figures 2 and 4 have been markedly changed to improve Readers’ experience.

Comment 2. The assignment of the TA transients to emitting and dark states seems sometimes arbitrary in absence of fs up-conversion experiments.

Response: The reviewer brings another valid point. While under this pandemic “slow down” we do not really have an access to up-conversion pump-probe setups, time-correlated single photon counting (TCSPC) studies allowed for unequivocal assignment of the transients corresponding to long-lived emissive states (with lifetimes exceeding about 200 ps). We outline the TCSPC findings in Table 1.

For short-lived transients, TA spectroscopy is still immensely informative regarding if they are emissive or not. Transient spectra that show negative signals in the regions where the compound should fluoresce, but does not absorb, allows for assigning them to stimulated emission (SE). Detecting SE, therefore, renders the transient emissive. Not observing SE, however, does not mean that the transient represents a dark state (strong TA can overlap with the SE signals and overwhelmingly cancels them). We added several sentences that bring these point across, and

revised the discussion of the results to clarify further the rationales for the assignments of the observed transients.

Comment 3. One important item of the present paper is the competition between fluorescence and intersystem crossing in the six nitro-derivatives. For this purpose, first of all the experimental derived k_f should be clearly reported in a table and compared with the calculated values to support the reliability of the calculations. As to the kISC and the eventual presence of significant population of the triplet state the authors completely rely on the calculations even if, in some points along the text, they admit that the calculations appear to overestimate kISC. In my opinion the measure of the triplet formation yield for the investigated nitro-compounds in different solvents is necessary to a reliable comparison.

Response: We agree with the reviewer's comment regarding the importance to the present k_f in a clear manner. Therefore, we summarized the experimentally obtained rate constants of radiative and non-radiative decays in the new Table 1.

Indeed, we rely on the theoretically calculated kISC. Experimentally measured ISC quantum yields will provide additional confirmations. Our lab housing the transient-absorption instrumentation, however, is still closed under the university social-distancing restrictions.

Nevertheless, because triplet formation is not the principal mode for non-radiative decay for most of the cases in this study, ISC quantum yields would not significantly impact the major conclusions for these nitro compounds.

All evidence show that except for two of the compounds in cyclohexane and one in DMSO, all modes of non-radiative deactivation are dominated by IC, which is a key finding of this study. Our mechanistic assignments, therefore, heavily rely on the experimentally obtained k_{nd} and we use the computed kISC to further confirm these interpretations.

Comment 4. The main purpose of this study would be to suggest structural design for nitro-aromatics leading to high fluorescent systems. However, in the Conclusions section it is not clear what are the strategies for obtaining fluorescent nitro derivatives. It seems that the only suggestion is to placing nitro groups on the phenyl substituents but I know many examples of nitrophenyl derivatives that show different emissive behaviour. Please better summarize your results.

Response: We thoroughly agree with the reviewer. Therefore, we dedicated a considerable part of the new "Discussion" section to the implications of the major findings from this study to the development of fluorescent nitro compounds and organic materials.

Reviewer 3.

The manuscript describes six bis-nitroterraphenylpyrrolopyrrole acceptor-donor-acceptor systems. The main result of this contribution is illustrating the understanding of the the large variability of the photophysical behavior of these systems as a function of solvent-environment, substituent positions. These systems are excellent models for the development of fluorophores where the nitro group plays an essential role and implies the presence of several photophysical channels. The model to study these effects correspond to a series of dihydropyrrolo-pyrroles systematically substituted with nitrophenyl groups. The introduction is excellent and gives a broad understanding of the effects of the NO₂ group in chromophores. Among the six molecules of this study, only the one with the NO₂ group in the ortho position with a single phenyl bridge is non-fluorescent. The emission of these systems undergoes red-shifting with increasing the solvent polarity and a reduction in the emission yield. Upon increasing bridge of these systems with a n additional phenyl group, the molecules have a relative increase in the Stokes shift. When the systems are placed in a rigid solid environment, there is an important increase in the emission yields and a reduction in the Stokes shifts. The study also includes TDDFT calculations that indicate that there are two types of pathways for the

locally excited states. One that forms a CT state and another one that forms a TICT-type (dark) state. Another important finding of this contribution is that from the electronic structure of these molecules, the SOC matrix elements and intersystem crossing rates are particularly small in comparison with nitrated polycyclic aromatic hydrocarbons where the isc rates can be as large as $10e13\text{ s}^{-1}$. I consider that this manuscript signifies an important contribution with clear results about an important topic. I recommend publication after the following minor issues are considered by the authors:

We would like to thank the reviewer for evaluating our work positively and for recognizing the importance of our work in the field of pi-conjugated molecular systems. Below we provide answers to the important points raised by the reviewer.

Comment 1. In the introduction the authors should consider mentioning the possibility of the effect of the nitro group dihedral angle with respect to the aromatic systems, where, for some nitroaromatics with highly twisted NO₂ groups, tend to undergo nitric oxygen dissociation.

Response: Indeed we omitted this deactivation channel in our introduction. Given that we had to shorten the introduction we added only corresponding sentence (accompanied by two references): 'In cases when NO₂ group is highly twisted (e.g. 9-nitroanthracene), the important pathway from S₁ corresponds to a photochemical reaction which generates nitrogen(II) oxide (NO·).'

Comment 2. -In the section about solid solvent medium, the possibility for formation of CT states in these media is not mentioned explicitly. I think the readers will be expecting that this topic is included or at least commented more explicitly in this section.

Response: The reviewer brings a valid point. We elaborated on the comments of how and increase in solvent polarity increase the extent of charge separation of the excited state. Also, we report the calculated dipole moments and how solvent polarity enhances them. The Lippert–Mataga–Ooshika analysis of the spectral data, which we added, provides further insights about these solvent effects. We also added a short discussion about the assignments of some of the TA features to the absorption bands of the radical ions of the pyrrolopyrrole and nitrophenyls, which is an addition supporting evidence for the CT character of the observed excited states.

Comment 3. I do suggest that in addition to Figure 6, the authors attempt to summarize their results in a graphical scheme or simple table which indicates increases and decreases in the non-radiative and radiative rate constants depending on the NO₂ position (and the reason, in terms of the respective state shifting), presence of an additional phenyl bridge, or the solvent environment, and use of a solid environment; otherwise it is a bit difficult to quickly find a specific effect for a given molecule.

Response: The new Table 1 summarizes the radiative and non-radiative rate constants obtained from the experimental and theoretical studies.

Comment 4. The same kind of comment would apply to the conclusions section. The manuscript is quite thorough in its discussions in the results section. A summary of the effects for each molecule should be found in the conclusions section.

Response: The reviewer brings another valid point. In the new Discussion" section we summarize the results and how different molecular features affect the fluorescence properties of the nitro compounds.

Sincerely yours

Daniel Gryko, Valentine Vullev and Hans Agren

REVIEWERS' COMMENTS:

Reviewer #3 (Remarks to the Author):

In my opinion the manuscript takes on an important and fundamental topic related to electronic structure of higher states and their interplay with the nitro group, one of the most important electron withdrawing substituent.

I recommend publication and include the following comments. At the end I include two additional minor issues (MINOR PENDING ISSUES) that the authors should take into account for the final version of the manuscript.

Reviewer 1 comments positively on the manuscript. I coincide with her/him that this contribution is relevant since it explores the possibility that nitro-substituted chromophores can become fluorescent under certain conditions. Reviewer 2 rejects the paper (or requires a new version) on the grounds that it needed major restructuring which, I believe has occurred.

The authors have included a new discussion about steric hindrance effects and make appropriate comparisons between the different nitro substitutions.

The authors have selected not to include new studies about the distribution of the molecular orbitals, and the density of states. I believe that the manuscript is already quite long and that the main results are presented well by the authors. The authors actually do include new elements on the discussion of the extension of the pi-conjugation in response to comment 6 of reviewer 1. The authors have included additional lines related to the efficiency of the coupling between the benzene and pyrrolo-pyrrole units, addressing the respective comment made by reviewer 1. More detailed comparisons between calculated and observed electronic transitions is included in the S.I.

Other minor corrections have been taken care by the authors. Also, new references with respect to the presence of isoenergetic triplet states was included.

Reviewer 1 required significant improvements on the computational analysis part. In response, the authors go to some length in explaining how they determined the structure and brightness of the S1-CT state (given that a S1-TICT state is the global minimum).

The authors have included in the response, calculations using the CAM-B3LYP, B3LYP and Wb97XD functionals in order to support the use of the long range corrected functionals in their calculations. Spectroscopic data has been summarized in the SI and in Tables 1 and 2 from the requirement of reviewer 1.

The authors have added element based on TCSPC and stimulated emission related to the assignment of transient absorption signals to emissive species. This touches on comment 2 by reviewer 2. I believe this was a valuable addition.

Importantly, the authors have added a new Discussion section with general trends, and summarizing the results of this study (all reviewers saw this as necessary).

The authors now include a brief mention of the nitric oxide photodissociation channel present in some nitroaromatics (this will allow readers interested in nitroaromatic photoreactivity to know about the presence of this process).

MINOR PENDING ISSUES

In regards to comment 14 by reviewer 1, I consider that the authors should at least try to comment on the relative accuracy of the predicted oscillator strengths through comparisons with the absorption cross sections observed for the different transitions of the different molecules (thereby validating the use of TDDFT calculations to ascertain the oscillator strengths).

The dihedral angles of the acceptor nitrophenyl groups with respect to the pyrrolo-pyrrole core appears to be a critical parameter for the couplings. I suggest including a table with these calculated values for the molecules of this study. The manuscript indicates Figures 1 and Supporting Figure 13 as illustrative of these angles, however, these figures only show molecular structures from where the dihedral angles trends are impossible to visualize.

Here is point-by-point response:

Reviewer 3

Comment 1. In regards to comment 14 by reviewer 1, I consider that the authors should at least try to comment on the relative accuracy of the predicted oscillator strengths through comparisons with the absorption cross sections observed for the different transitions of the different molecules (thereby validating the use of TDDFT calculations to ascertain the oscillator strengths).

Response: as requested by Referee, we calculated the absorption profiles by using a Gaussian model [S.I. Gorelsky, A.B.P. Lever, J. Organomet. Chem. 2001, 635, 187–196]. The values of extinction coefficient calculated at the absorption maximum (ϵ_{theor}) are compared with the experimental data (ϵ_{exp}) in the Table below. One can see, that DFT calculations generally overestimates the experimental values of extinction coefficient at the absorption maxima (by 2 times approximately). But quantitatively, calculated and experimental data correlate perfectly showing the growing extinction coefficient within series of **1o,1m,1p** and **2o,2m,2p**. Also, calculations reproduce the higher absorption intensity by the series 2 species comparing to series 1 species (**2o** absorbs stronger than **1o**, etc.). Therefore, we can validate the usage of TDDFT method for correct prediction of absorption intensity by studied species.

molecule	ϵ_{exp} at first absorption maximum, $\text{M}^{-1} \text{cm}^{-1}$	ϵ_{theor} at $E(S_0-S_1)$, $\text{M}^{-1} \text{cm}^{-1}$
1o	6.2×10^3	1.7×10^4
1m	3.7×10^4	2.1×10^4
1p	4.7×10^4	1.1×10^5
2o	3.6×10^4	9.6×10^4
2m	4.7×10^4	1.5×10^5
2p	5.0×10^4	1.6×10^5

Comment 2. The dihedral angles of the acceptor nitrophenyl groups with respect to the pyrrolo-pyrrole core appears to be a critical parameter for the couplings. I suggest including a table with these calculated values for the molecules of this study. The manuscript indicates Figures 1 and Supporting Figure 13 as illustrative of these angles, however, these figures only show molecular structures from where the dihedral angles trends are impossible to visualize.

Response: we prepared the additional table containing the dihedral angles between pyrrolo-pyrrole core and nitrophenyl groups and introduced it into the supplementary materials file.

	θ_1 , deg.			θ_2 , deg.			θ_3 , deg.			θ_4 , deg.		
	S ₀	S _{1CT}	S _{1TICT}	S ₀	S _{1CT}	S _{1TICT}	S ₀	S _{1CT}	S _{1TICT}	S ₀	S _{1CT}	S _{1TICT}
1o	54.5	42.7	54.7	54.5	42.7	95.4	-	-	-	-	-	-
1m	38.0	21.5	29.3	38.0	21.5	66.6	-	-	-	-	-	-
1p	36.0	20.9	29.9	36.0	20.9	87.5	-	-	-	-	-	-
2o	39.3	21.4	33.1	39.3	21.4	32.1	49.5	42.1	89.7	49.5	42.1	54.8
2m	38.3	20.5	29.7	38.3	20.6	38.0	35.6	28.1	36.0	35.6	28.2	39.3
2p	37.9	20.8	93.5	37.9	20.8	28.8	35.0	26.3	30.7	35.0	26.5	37.4